# A digital twin for parallel liquid-state nuclear magnetic resonance spectroscopy
Mengjia He [1], Dilara Faderl[1], Neil MacKinnon [1] ✉, Yen-Tse Cheng[1], Dominique Buyens[1], Mazin Jouda [1], Burkhard Luy[2,3] & Jan G. Korvink [1] ✉

One approach to increasing nuclear magnetic resonance measurement sample throughput is to implement multiple, independent detection sites. However, the presence of radio frequency interference poses a challenge in multi-detector systems, particularly in unshielded coil arrays lacking sufficient electrical isolation. This issue can lead to unwanted coupling of inductive coils, resulting in excitation pulse interference and signal transfer among multiple detection sites. Here we propose a theoretical framework that combines electromagnetic simulation with spin-dynamic calculations. This framework enables the evaluation of coil coupling effects, the design of parallel pulse sequences to mitigate inter-channel coupling, and the separation of composite free induction decays obtained from multiple detectors. The parallel pulse compensation scheme was validated by a 2-channel parallel spectroscopy experiment. These results provide valuable insights for the design of parallel nuclear magnetic resonance hardware and for exploring the limits of parallelization capacity within a fixed magnet system.

Nuclear magnetic resonance (NMR) spectroscopy has been widely used to reveal detailed molecular structure and structure-function relationships. Whereas NMR technology is routinely applied to analyze large numbers of samples or to monitor dynamic biochemistry processes, high-throughput capability has remained elusive. A straightforward way to increase throughput is to detect multiple samples simultaneously via multiple channels, broadly termed parallel NMR[1]. Several previous studies utilize multiple radio frequency (RF) coils for parallel excitation and reception[1–3]. Two broad approaches have been explored with regard to hardware design: The first method combines all coils into a single circuit with a shared tuning and matching circuit. It separates signals using a pulsed gradient field, which reduces electronic components but collects noise from the entire coil array. This configuration is limited to cases[1,4] which have a relatively high signal-to-noise ratio (SNR). The second design deploys each coil with its own tuning and matching circuit and receiver chain[2,5], where each sample can be excited and detected without the use of gradient fields as a means to achieve effective parallelization. The primary obstacle is achieving exceptional electrical isolation[6] among parallel channels, in other words, the inter-coil coupling should be sufficiently weak. The method of suppressing coupling was recently discussed in a sample-centered shimming experiment[7], where the coil geometry was optimized to minimize the coupling effect. However,

the RF shielding design is practically limited by space and probe complexity, so the coupling cannot be completely eliminated. How to handle the unavoidable coupling remains as an open question. Our work explores software-based approaches to track the inter-channel coupling issue, both in the excitation stage (i.e., parallel pulse sequence design), and the reception stage (composite signal splitting).

In the realm of parallel detection, the incorporation of multiple receive coils also stands as a milestone within the MRI community[8,9]. This approach, known as 'parallel imaging,' has garnered approval for its capacity to enhance SNR[10] and reduce acquisition time[11]. The evolution of parallel imaging led to the utilization of multiple transmit coils in the context of 'parallel transmission' (pTx)[12]. This application aimed at improving RF excitation, including enhancements in $B_1$ homogeneity[13], minimization of the specific energy absorption rate[14], and even RF shimming[15]. Addressing the spatial sensitivity of each transmit element, the RF pulse underwent optimization to reduce the duration in spatially selective multidimensional excitation[16]. Given the dependence of the transmit $B_1$ field on the experimental platform, subject-specific pulse optimization involved exploiting $B_0$ and $B_1$ maps as prior knowledge[17–19]. To circumvent field measurement and data processing, the concept of universal pulses emerged, demonstrating robustness against subject-

[1]Institute of Microstructure Technology, Karlsruhe Institute of Technology, Eggenstein-Leopoldshafen, Germany. [2]Institute for Biological Interfaces 4—Magnetic Resonance, Karlsruhe Institute of Technology, Eggenstein-Leopoldshafen, Germany. [3]Institute of Organic Chemistry, Karlsruhe Institute of Technology, Karlsruhe, Germany. ✉e-mail: neil.mackinnon@kit.edu; jan.korvink@kit.edu

dependent field inhomogeneity[20,21]. These universal pulses, designed based on a database of $B$-field maps from representative subjects, have been further refined using the GRAPE algorithm. For instance, the simultaneous optimization of RF and gradient waveforms can address $B_1$ heterogeneity[22] and off-resonance effects[23]. However, in the context of parallel NMR, the parallel concept aims to increase throughput, with each coil possessing a confined $B_1$ field to excite a specific sample. The overarching objective is independent operation for each channel, making the coupling-induced 'parallel transmission' a hardware non-ideality that needs resolution. This is distinct from strongly coupled pTx in MRI, where pTx is intentionally employed to enhance excitation.

The mutual coupling within parallel MRI transmit arrays has been extensively studied, prompting the development of diverse decoupling strategies. These approaches including employing decoupling capacitance[24], passive decoupling network[25,26], or preamplifier decoupling through impedance mismatch[27]. A recent review article has summarized these decoupling methods[28]. While the preamplifier decoupling needs substantial effort on system integration, active decoupling found more interest. One example entails controlling the currents in eight-channel transverse electromagnetic transmit coils to attain independent transmit sensitivities, with control coefficients designed based on the impedance matrix of the coil array[29,30]. This active decoupling concept might be combined with the tailored RF pulses which have become the principal approach for manipulating nuclei spin. In this context, we explore parallel excitation by designing cooperative pulses aimed at mitigating inter-channel coupling, while avoiding the $B$ field mappings. The overall effect of the cooperative pulse would be to achieve excitation, equivalent to that produced by each channel in the absence of coupling, and hence a coupling model for parallel excitation is crucial for this purpose.

As such, the approach is an extension to the design of non-identical, but cooperatively acting pulses as previously introduced for related heteronuclear Hartmann-Hahn sequences[31], cooperativity for added scans[32], and consecutive scans[33].

In signal reception, understanding the impact of coupling on signal transfer is crucial. We need to determine how multiple free induction decays (FIDs) combine to generate detected signals, necessitating the development of a specific coupling model for the reception stage. In addition, we noticed that processing multi-channel NMR signals falls within the realm of handling composite signals from multiple detectors. In this domain, blind source separation (BSS) stands out as an exceptional model with diverse applications, including speech recognition[34], image processing[35], and biomedical signal processing[36]. BSS methods, under appropriate assumptions such as the independent sources condition[37], not only identifies signals from multiple sources, but also estimates the mixing matrix, which reveals the inter-detector coupling information. Therefore, the BSS method will serve as a complementary approach for the developed model in order to collaboratively uncover the inter-channel coupling.

In this work, an NMR simulation model was built to investigate the effects of inter-channel coupling in the case of a parallel detector configuration. This model accounts for the coil coupling-induced excitation distortion and FID composition. For these purposes, an electromagnetic simulation module was first built to calculate the $B(t)$ field ($B_0$, $B_1$) within the multiple, parallel samples, the spin-dynamic calculations were performed by importing the $B(t)$ field data and solving the Liouville-von Neumann equation[38]. Based on this model, a parallel pulse compensation scheme was put forward to address the RF coupling. Furthermore, we utilized the calculated coupling matrix to virtually separate the simulated composite signals and employed the BSS method to split both simulated and experimental signals. This enables the recovery of detector-specific signals and the identification of coupling. Finally, the pulse compensation scheme was validated through a parallel spectroscopy experiment using two channels. These results will help explore the limit of designing parallel NMR probes and experiments in a fixed volume, hence increasing sample throughput.

## Results

The ultimate objective of this simulation study is to explore methods to handle RF coupling in parallel NMR, i.e., parallel pulse sequence optimization, and parallel signal decomposition. The initial step concerns developing a model to reveal how coupling influences parallel NMR, including an electromagnetic simulation to predict the resulting magnetic field, and a means to quantify coupling, as well as the spin-dynamic calculation using the simulated magnetic fields in order to reveal coupling effects on the spin.

Firstly, we constructed a simulation workflow for single-channel NMR experiments, inspired by a simulation methodology for predicting MR noise[39] (Fig. 1). This workflow includes three parts: magnetic field acquisition, spin-dynamic simulation, and pulse sequence optimization. In the first part, we use COMSOL multi-physics to calculate the static magnetic field ($B_0$) and RF field ($B_1$) inside a sample, with well-defined coil geometry and electrical parameters (e.g. permittivity and permeability). The calculated fields were imported into the simulation package Spinach[40] for spin dynamics calculations in part 2. Finally, part 3 focuses on pulse optimization considering the magnetic fields predicted from part 1.

This workflow was then adapted for a parallel NMR scenario, where the electromagnetic (EM) simulation considers a detector array, and the spin-dynamic calculation accommodates the spin evolution of multiple samples. To illustrate the framework without losing generality, we selected water samples and solenoid RF coils for the EM simulation, and an 11.74 T magnet for the primary magnetic field.

### Modeling of parallel NMR

**EM simulation.** In parallel NMR experiments, multiple RF coils and samples are arranged as an array inside the magnet bore, where the coils couple with each other via an inductive-coupling effect (inter-coil coupling). In addition, the magnetic field from a local coil can spill into the

---

**Fig. 1 | A multi-physics simulation workflow.** The workflow consists of three stages: magnetic field acquisition, spin dynamics calculation, and pulse sequence optimization. Every stage includes several sub-steps that are summarized in a box below and highlighted in an image above. DC denotes a direct current, RF denotes radio frequency, $B_0$ is the static magnetic field, and $B_1$ is the radio frequency magnetic field.

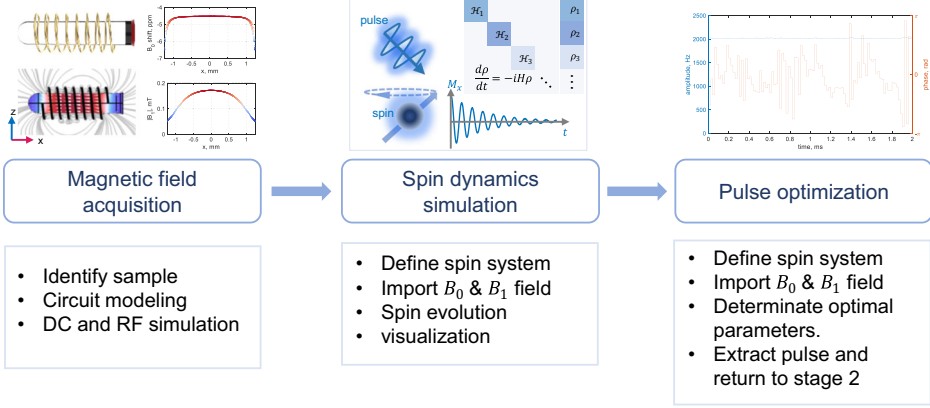

**Fig. 2 | Schematic view of the parallel detector.**
**a** One channel of the parallel detector, the sample is inserted into a solenoid coil, which is tuned and matched (T& M), and connected to the transmitter (Tx) or receiver (Rx) via a switch. RF denotes radio frequency, PA denotes the power amplifier in the Tx path, LNA labels the low-noise amplifier in the Rx path. **b** The geometry of the 4-coil array simulation in COMSOL, the 4 samples are shown in blue. **c** Circuit model of the parallel detector with S-parameters, the amplifiers are substituted by matched loads assuming that all the amplifiers are tuned and matched to the cable's characteristic impedance (50Ω). Free induction decays are denoted by FID. The symbols $a$ and $b$ indicate the forward and reverse waves respectively. $S_C$, $S_M$, $S_{CM}$ denote the S-matrices of the coil array, the T& M network, and their combination, respectively.

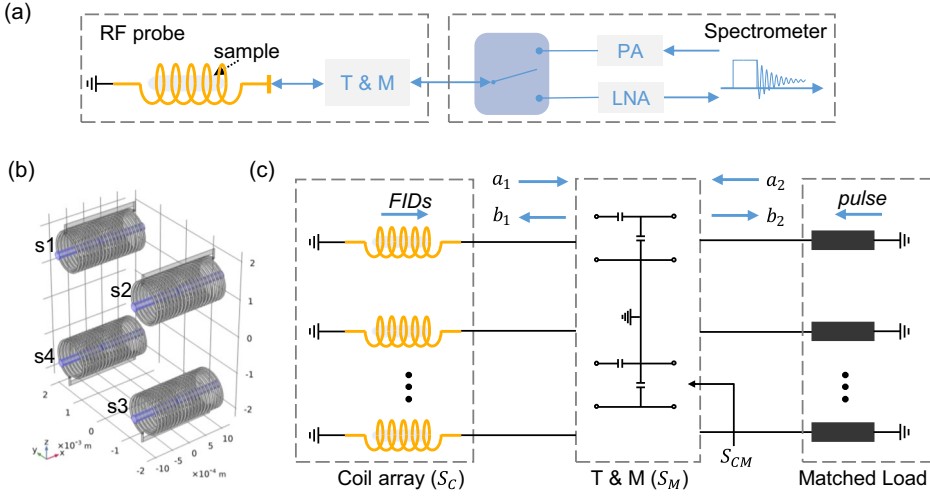

external space and affect neighboring samples (coil-sample coupling), especially in unshielded coil arrays. To quantify the coupling and its impact on the parallel system performance, the RF coupling chain was modeled with the S-parameters, which relates the voltage waves incident on the RF ports to those reflected from the ports[41]. The $B_1$ field was calculated by full-wave simulation with the calculated currents on the coils (Supplementary Note 1).

In Fig. 2a, a schematic of a single-channel RF probe is depicted, and a parallel NMR detector consists of multiple repetitions of this probe. Each channel is equipped with an individual tuning and matching (T&M) circuit, the T&Mcircuit is connected to the spectrometer, where the RF switch alternates between the Tx and Rx modes for excitation and reception, respectively. In Fig. 2c, we depict an RF chain built for modeling the coil array and T&Mnetwork. The amplifiers are substituted by matched loads, assuming that all the amplifiers are tuned and matched to the cable's characteristic impedance. We use S-parameters referenced to 50 Ω to combine each part, and the forward and reverse traveling waves are indicated as $a$ and $b$, respectively. These two waves can incorporate externally induced signals, including the FID and RF pulse.

To illustrate the RF chain, we first built a 4-solenoid array model together with cylindrical water samples in COMSOL, as shown in Fig. 2b, the feed ports connected to the T&Mnetwork were substituted by lumped ports with 50 Ω impedance. The RF simulation was carried out using the COMSOL 'emw' module, employing a frequency domain solver that solved the electric field wave equation through the finite element method. The frequency was set at 500 MHz, corresponding to the $^1$H Larmor frequency under an 11.74 T magnetic field. The simulation provides the coil array S matrix, denoted as $S_C$, which correlates the voltage waves incident on the coils with those reflected from the coils. Additionally, the simulation generates the prototype $B_1$ field under unit excitation (1 V). The T&Mnetwork was virtually designed, with each coil individually tuned and matched based on simulated $S_c$. In cases of very weak coupling ($S_{C,21} < -60$dB), capacitance sets were calculated using diagonal elements of $S_C$, achieving reflections below -20 dB. However, when coupling increases and leads to undesirable mismatches ($S_{CM,11} > -20$dB), we employed the Nelder-Mead simplex algorithm[42] for capacitance optimization (Supplementary Note 2). In this process, all channels shared the same capacitance sets, considering the symmetric geometry of the coil array. The calculated capacitances were then integrated into the coil array, resulting in an RF chain characterized by S-parameters.

The $B_1$ field at a specific position under parallel excitation can be expressed as the linear superposition of the field values obtained from separate excitation of individual coils[43], represented by $B_1 = \sum_m B_{1m}$. As RF coils are linear components, $B_{1m}$ can be expressed as $c_m I_m$, where $I_m$ represents the actual current flowing through the $m$-th coil, calculated using

the RF chain model. By performing a complete port sweep in COMSOL, we obtain $N$ values of $B_1$ and $N$ combinations of port currents, allowing for the determination of the linear coefficients $c_m$. Consequently, we could simulate the $B_1$ field under a given excitation $p$ using the equation

$$B_1 = \mathbf{T} \cdot \mathbf{F} \cdot \mathbf{p}. \tag{1}$$

Here, **F** transfers the excitation **p** to the coil current, and **T** converts the current to the $B_1$ field (Supplementary Note 3). As we iterate through each position, the full-wave simulation results are reserved, including the phase information. Using this approach, we simulated the $B_1$ field for a 4-solenoid array, assuming only the first channel was excited in Fig. 2b. It's important to note that we apply an incident voltage $p$ at the T&Mnetwork terminal, assuming that the amplifier precisely delivers the required power to the RF probe. Therefore, the amplifier should operate within its linear range, and caution should be exercised when considering very high pulse powers[44]. In addition, the RF simulation conducted in the frequency domain yields steady-state results, so the transient response of a finite-Q circuit was neglected. This matter was addressed concerning simulating the transient response[45] and measuring the impulse response function[46].

To estimate the homogeneity of the static $B_0$ field, we replicated the geometry used in the RF simulation within the COMSOL 'mfnc' module. The sample susceptibility was set to $-9.035 \times 10^{-6}$, and the static field was determined based on Gauss' law $\nabla B = 0$ within an 11.74 T background field.

**Modeling the excitation stage.** Figure 3a illustrates the signal flow during parallel excitation. The first stage involves calculating the $B_1$ field using the provided pulse, supposing the synchronous pulse in each channel. The parallel RF pulses are amplified to a predetermined power level and delivered to the terminals of the T&Mnetwork. At the T&Mnetwork port, the excitation pulse **p** contributes to the forward waves **a₂**, as depicted in Fig. 2c. The $B_1$ field is computed using Eq. (1), enabling the determination of the field distribution across the entire space.

In the second stage, we address the spin evolution for multiple samples by incorporating them into a composite system that encompasses all spin isotopes and considers spin interactions. Mathematically, the system Hamiltonian is constructed by organizing the sub-Hamiltonians into blocks of a block-diagonal matrix, where each block corresponds to one sample. The corresponding spin state vectors are also assembled to align with the Hamiltonian. In Spinach, when defining the spin system, interactions between different samples are neglected under the assumption that sample-sample coupling, e.g. via radiation damping, can be disregarded (Supplementary Note 4). Spin operators are calculated and applied separately to

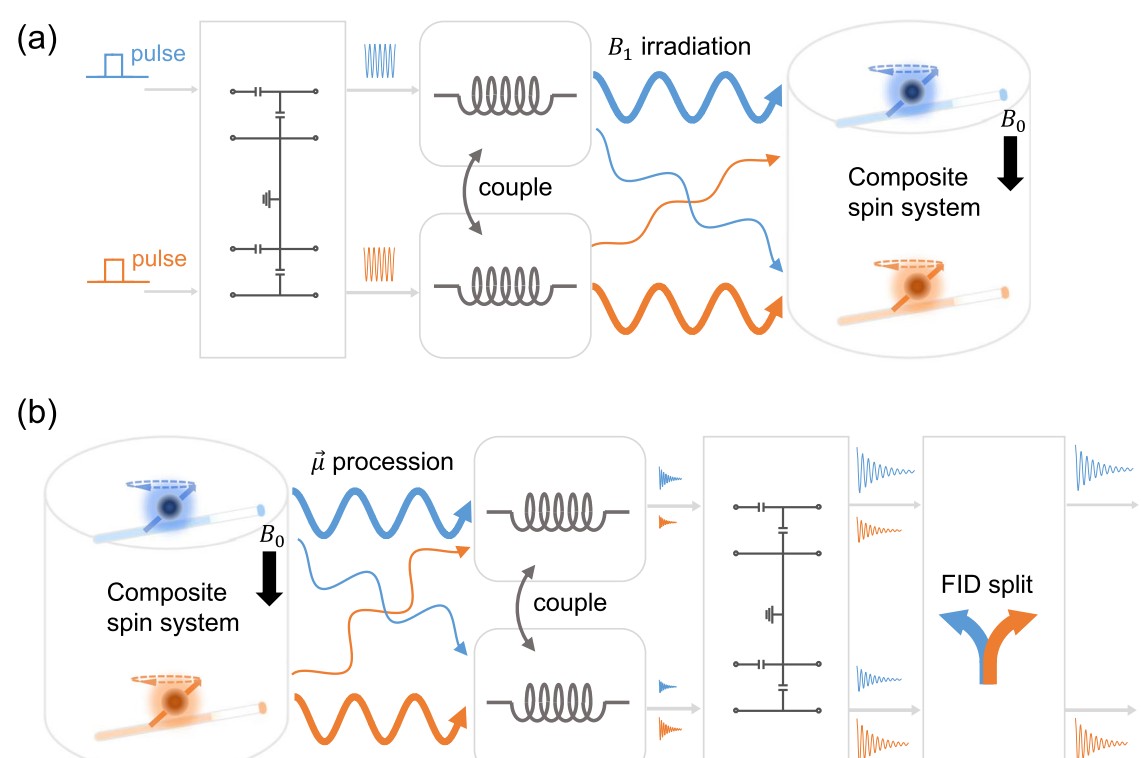

**Fig. 3 | The framework for modeling the parallel nuclear magnetic resonance experiments. a** The excitation stage, the parallel pulse are processed for calculating the coupled $B_1$ field, and a composite spin system is built in Spinach to accommodate the spin-dynamic calculation of multiple samples in the probe, all channels are excited with synchronous pulse sequences. **b** The reception stage, the magnetization is recorded by the radio frequency coils, suffering from coupling, and then delivered to a post-processing module for signal decomposition. The FID is free induction decay, $\mu$ is the magnetic moment, $B_0$ is the static magnetic field and $B_1$ is the radio frequency magnetic field.

spins from different samples. Therefore, the spins from multiple samples are individually excited, evolved, and detected, despite being within a single composite spin system.

This workflow also enables heteronuclear simulation, which applies to polarization transfer and high-dimensional NMR experiments. The relevant $B_1$ fields would only need to be calculated for each desired nucleus.

**Modeling the reception stage.** Figure 3b shows the signal flow for parallel reception, each channel acquires the FIDs synchronously, i.e., all channels are fixed with the same sweeping frequency and acquisition period. The spin evolution was conducted in this period, giving the trajectories of the spin state, which is projected to the vector representing the coil to calculate the normalized FID, i.e, $\mathbf{m} \cdot \mathbf{b}^- = \langle\text{coil}|\rho\rangle$[47]. The real FID was calculated according to the reciprocity principle[48],

$$f = \int i\omega_0 |M| |B_1^-| e^{i\phi} \mathbf{m} \cdot \mathbf{b}^- \, dV. \tag{2}$$

The relaxation effect was neglected, and the magnetization amplitude $|M|$ was fixed at 1. The receiving field $|B_1^-| e^{i\phi} = B_x - iB_y$ was calculated using the same method as the transmit $B_1^+$, assuming a unit current through a specific coil. Signal calculations for the received signal utilized the RF chain depicted in Fig. 2c. Within the RF chain, the FID contributed to the open-circuit voltage at the coil terminals, a total gain was determined to transfer the FID to the T&Mnetwork terminal (Eq. S10). Ultimately, the signal at the T&Mnetwork output formed the input for the signal decomposition module, further details on signal decomposition will be provided in the subsequent section.

The RF coupling between heteronuclear channels is ignored because they are tuned to isolated frequencies. The acquisition for each nucleus shall adapt to an independent RF chain and a given coupling strength.

**Parallel pulse compensation scheme**
In the parallel NMR experiments, multiple channels deliver simultaneous RF pulses. Poor electrical isolation can cause coil coupling, i.e., the $B_1$ field generated at one channel is sensed by neighboring samples, resulting in undesired manipulations of their spin states. In this sense, normal pulse calibration for a single sample fails in the parallel case, since the $B_1$ field distortion comes from both the individual coil imperfection as well as from the pulse leakage from adjacent channels. To overcome imperfections in single coils, optimal control methods have been extensively employed to generate composite pulses, which exhibit high-performance transfer efficiency against field inhomogeneity and resonance offset[49–52], and have been used to explore physical limits of excitation and inversion pulses[53–55]. In the parallel case, the advantages of optimal parallel transmission for compensation of patient-induced $B_1$ inhomogeneity and reducing RF power in MRI have been demonstrated[8], and the cooperative pulses for multiple-scan spectroscopy[32] contribute similar inspiration. Utilizing the obtained $B$-map database, the concurrent optimization of pulses across multiple channels enhances robustness against load-induced $B$-field inhomogeneity, thereby improving single-sample excitation in MRI. However, in parallel NMR involving multiple and potentially distinct samples, the pulse applied to each channel must prioritize specific sample excitation while guarding against coupled radiation from neighboring channels. If the inter-channel coupling details are known, the simultaneous optimization can be transformed into channel-specific compensation. This approach may grant the signal-channel pulse greater flexibility in addressing field inhomogeneity and spin-spin interactions[56–58]. Here, we combine single-channel pulse optimization with the coupling information provided by RF modeling, to design the parallel pulse sequences.

From parallel detector modeling, we correlate the $B_1$ field with the pulse sequence through two matrices, $B_1 = \mathbf{T} \cdot \mathbf{F} \cdot \mathbf{p}$, where $\mathbf{p}$ is the excitation at the T&Mnetwork terminal ($a_2$ in Fig. 2c). The $\mathbf{F}$ and $\mathbf{T}$ matrix indicate a two-

step linear supercombination. Specifically, the **F** transforms the excitation pulse to coil currents, so it represents a circuit-level combination. This combination is position-independent and convenient to be canceled out by pulse compensation. The **T** matrix transforms the coil currents to the $B_1$ field, so it represents a field-level combination, which contributes to the direct field spill-over effect from one channel to another. In this way, we separate the mutual coupling into circuit-level combination and field-level combination. Generally, the RF coils including striplines and solenoids confine the $B_1$ field at their center, so the outgoing radiation becomes very weak and makes a minimal contribution. Consequently, in weak coupling cases, the simultaneous optimization of parallel pulses incorporating the position-dependent **T** matrix can be avoided. The parallel pulse compensation is aimed to cancel out circuit-level combination.

Prior to the pulse compensation, we examine the impact of coupling on pulse efficiency, specifically the decrease in excitation fidelity under a range of coupling strengths. In Fig. 4a, by subjecting one $^1$H spin to a single 90° pulse in both channels, we observed a remarkable decrease in fidelity as the coupling strength increased, indicated by the close proximity of the two solenoid coils. Notably, we assigned the proton chemical shifts as -2 ppm for channel-1 and 5 ppm for channel-2, respectively. Consequently, a slight discrepancy between the two lines arose as a result of the distinct impact of the coupled pulse on each nucleus. Similarly, Fig. 4b presents the excitation fidelity for the optimized control pulse, which was specifically designed to exhibit stability in the presence of resonance offset and $B_1$ field inhomogeneity. In the absence of coupling, the optimized control pulse achieved near-perfect fidelity. However, as the coupling strength increased, a remarkable decline in fidelity was observed, mirroring the behavior of the hard pulse case. Notably, Fig. 4b demonstrates that the fidelity degradation resulting from RF coupling exhibits variations between the two channels. This discrepancy is attributed to the disorganized pulse shape which is distinct for the two channels. In addition, the adverse effects of coupling are heightened with increasing pulse power, the functionality of the 10 kHz pulse is entirely compromised when operating in parallel with a coupling strength of 0.24, as detailed in Supplementary Fig. S8.

Due to its ability to precisely manipulate nuclear spins, the optimal control pulse offers exceptional performance. However, the RF coupling disrupts its waveform by inducing additional currents on the coil. Consequently, the pulse becomes less robust to resonance offset and field inhomogeneities, resulting in a decrease in transfer efficiency. To mitigate RF coupling through pulse compensation, the key concept is to create a cooperative pulse that integrates the optimal control pulse with a compensation term (refer to Fig. 5a). The cooperative pulse aims to exert an

equivalent effect on the spin as the optimal control pulse does during individual excitation. This approach can be outlined in the following steps:

1. Run optimal control for each channel individually, generating pulse sequence $\mathbf{p_0}$.
2. Assume a compensation term $\delta_i$ for each channel, to satisfy the elimination condition, for which the following equation is established

$$\mathbf{F} \cdot \begin{pmatrix} p_{01} + \delta_1 \\ p_{02} + \delta_2 \\ \vdots \\ p_{0n} + \delta_n \end{pmatrix} = \begin{pmatrix} F_{11} & & & \\ & F_{22} & & \\ & & \ddots & \\ & & & F_{nn} \end{pmatrix} \begin{pmatrix} p_{01} \\ p_{02} \\ \vdots \\ p_{0n} \end{pmatrix}. \quad (3)$$

3. Determine the cooperative pulse $\mathbf{p} = \mathbf{p_0} + \boldsymbol{\delta}$ by solving Eq. (3).

To give an example, in the first step, the single-channel optimal control considered transferring one $^1$H spin from $I_z$ to $I_x$, accounting for resonance offset ( ± 2.5kHz bandwidth) and RF inhomogeneity determined by the imported $B_1$ field. The $B_1$ field, calculated at a reasonable excitation power, generated an average RF amplitude of 2kHz for $^1$H. In the second step, the compensation term $\boldsymbol{\delta}$ was added to $\mathbf{p_0}$ to cancel the coupling effect, enabling the generation of coil currents consistent with individual excitation of the optimal control pulse. Fig. 5b illustrates the normalized excitation waveform for two-channel pulse compensation, displaying pulse sequences in terms of amplitudes and phase with 100 time slices and a duration of 2ms. Each element of $\mathbf{p_0}$ had a fixed amplitude of 1, whereas the amplitude and phase of $\mathbf{p}$ changed over time. Fig. 5

(c) presents the simulated transfer fidelity under three excitation conditions: applying the optimal control pulse individually without coupling ($\mathbf{p_0}$ - no couple), the optimal control pulse in parallel with coupling ($\mathbf{p_0}$ - coupled), and the cooperative pulse with coupling ($\mathbf{p}$ - coupled). When applying $\mathbf{p_0}$ in parallel, the coupling introduced $B_1$ field distortion and led to a drop in transfer fidelity. However, the compensation of $\mathbf{p}$ accounted for coupling and achieved the same efficiency as individual excitation of $\mathbf{p_0}$. It's important to note that this drop varied depending on the resonance offset and local $B_1$ field. Additionally, due to the step-by-step optimization process of the initial guess for the optimal control pulse, which contained randomness, the efficiency drop could differ between the two channels, as observed in Fig. 4b.

Under a wide range of coupling strengths, as long as Eq. (3) remains solvable, a cooperative pulse can achieve comparable transfer efficiency relative to the case in which no coupling exists and optimal control pulses

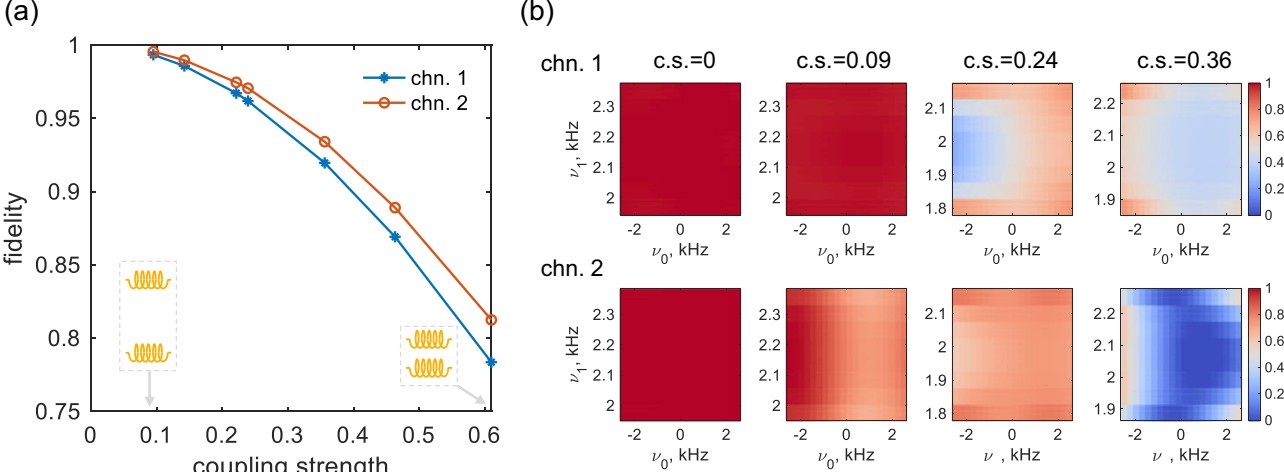

(a)   (b)

**Fig. 4 | The excitation fidelity $\langle \rho | I_x \rangle$ calculated for parallel pulse execution.** **a** Fidelity of the parallel hard pulse plotted against coupling strengths (c.s. $= |F_{12}/F_{11}|$). Each pulse has a 90° flip angle and lasts for 20 μs. **b** Fidelity of optimal control pulses applied in parallel under selected coupling strengths, $v_0$ and $v_1$

represent the resonance offset and radio frequency amplitudes, respectively. The radio frequency amplitude is determined based on the local $B_1$ field, considering a fixed excitation power, resulting in slight variations of its range as the coupling strength is modified. The two channels are indicated as chn. 1 and chn. 2.

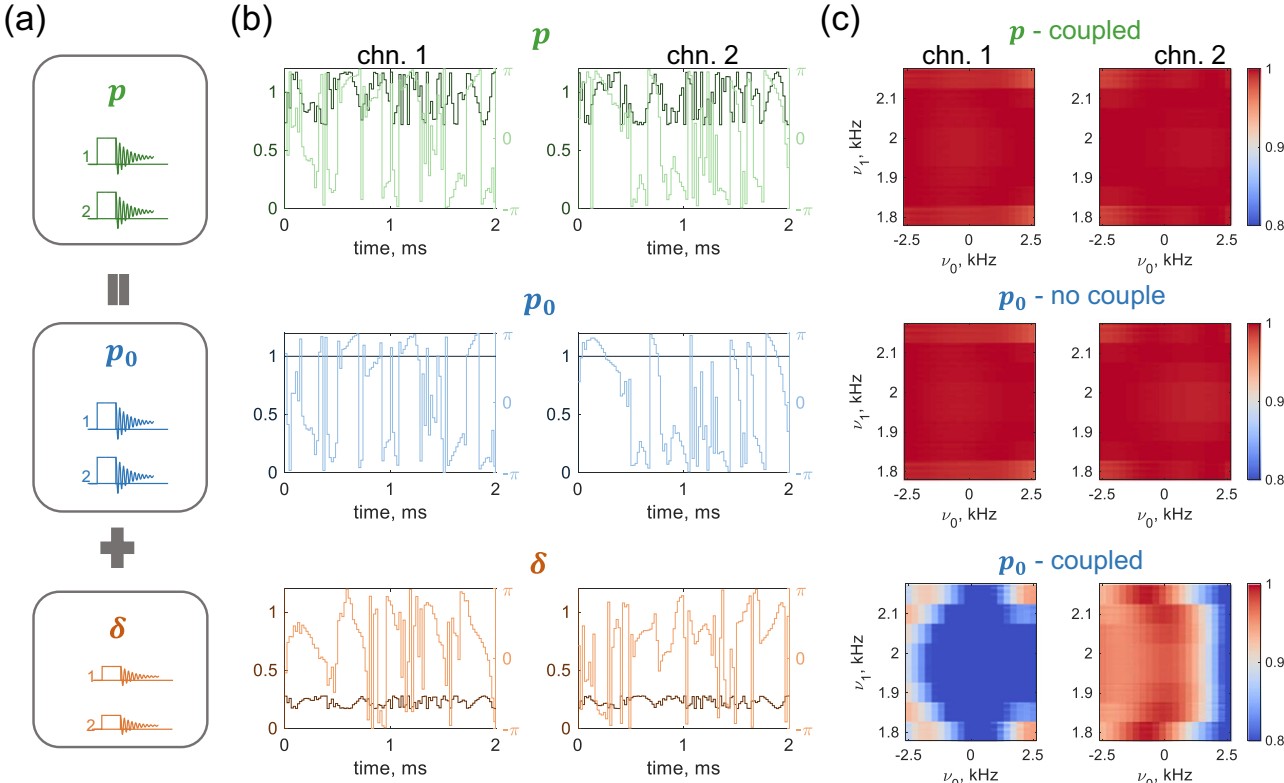

**Fig. 5 | Parallel pulse compensation. a** Schematic of pulse compensation. A cooperative pulse is obtained by combining the optimal control pulse with a compensation term. **b** Waveform at the tuning and matching network input for a cooperative pulse **p**, optimal control pulse **p₀**, and a compensation term **δ**. The dark-colored lines (left axis) represent the amplitude. The constant-amplitude optimal control pulse has the nominal radio frequency amplitude of 2kHz, and the light-colored lines (right axis) represent the corresponding phase. **c** Transfer fidelity when applying the cooperative pulse (**p**) with coupling, and the optimal control pulse (**p₀**) without and with coupling. The coil geometry, spin system, and transfer task are the same as those in Fig.4, and the coupling strength c.s. = 0.24. The two channels are indicated as chn. 1 and chn. 2. in (b) and (c). The $v_0$ and $v_1$ represent the resonance offset and radio frequency amplitude, respectively.

apply on isolated channels. The cooperative pulses maintain over 96% mean fidelity with coupling strengths equivalent to 0.09, 0.24, and 0.36 of 2 solenoid coils, provided in Supplementary Fig. S9. As a portion of the cooperative pulse energy is allocated to counteract coupling, its applicability may be constrained by available power, particularly in cases of strong coupling.

**Parallel signal decomposition scheme**

Coil coupling not only contributes to pulsed current leakage during the excitation stage but also causes FID cross-talk[3] in the reception stage. Here we develop a post-processing method to split the composite signal at the output of the T&Mnetwork (Fig. 3b).

Consider synchronous parallel acquisition for $n$ homonuclear channels. In this scenario, each coil records the FID from all samples, i.e., $s_i = f_{ii} + \sum_{j \neq i} f_{ij}$, where $f_{ij}$ is the FID on channel $i$ from sample $j$. Eq. (2) indicates that each FID component is determined by the spin magnetization as well as the $B_1^-$ field. When the $j$-th sample obtains uniform magnetization, for example, $M_j = M_x$, the coupled signal $f_{ij}$ could be given as $R_{ij}f_{jj}$, where $R_{ij}$ is the ratio of sensitivity. In this way, we reduce the unknowns and summarize that the received signal **x** can be expressed as (Supplementary Note 6):

$$\mathbf{x} = \mathbf{G}' \cdot \mathbf{f} + \mathbf{n}' \qquad (4)$$

where **f** is the signal vector consisting of the primary FIDs, i.e., $\mathbf{f} = (f_{11}, f_{22}, ..., f_{nn})^T$, and $\mathbf{n}'$ is the coupled noise, $\mathbf{G}'$ is a modified coupling matrix, in which $G'_{ii} = G_{ii}$, $G'_{ij} = G_{ij} + G_{ii}R_{ij}$. The **G** is the reception gain and the $R_{ij}$ is the sensitivity ratio between the coil $i$ and the coil $j$ regarding the sample $j$. Hence, the primary FIDs can be recovered provided the

coupling matrix is known. In a digital twin, the coupling matrix is calculated through EM simulation, in an experimental scenario, Eq. (4) could be solved by BSS[59,60], these two aspects are discussed below.

Figure 6 shows the signal decomposition results for a 4-detector, 8-channel array, where the pairs of ¹H and ¹³C channels were simultaneously excited and received, assuming perfect isolation between heteronuclear channels. For simplicity, only one resonance was simulated for each channel and was assigned a different chemical shift per channel. The $B_1$ fields were simulated for ¹H and ¹³C using the electrical parameters of water, i.e., assuming an aqueous sample. Figure 6a, c displays the coupled signal with interference. The interfering signals are relatively lower in the ¹³C channel due to a smaller coupling strength at the lower frequency. Figure 6b, d gives the decomposed signals, demonstrating complete splitting of the desired signals from the cross-talk component. As the modified coupling matrix originates from reception gain $G$ and sensitivity ratio, the decomposition error was simulated regarding the random shifts of $G$ and $B_1$, see Supplementary Note 7.

Note that Eq. (4) expresses the received signals as a linear combination of the FIDs providing the reception gain is time-invariant, indicating that signal decomposition is a typical BSS problem under instantaneous linear mixtures[60]. First, the source signals from multiple samples are statistically independent, and second, we suppose they have different spectral content. The second-order blind identification (SOBI) algorithm[61] is an exceptional and robust method and suitable in this case. This method is based on jointly diagonalizing a set of covariance matrices, which could be adapted to difficult contexts such as adverse SNR and sources with little spectral difference. In Fig. 7, we use the SOBI algorithm to decompose ¹H spectra from a two-channel array, the original spectra and split spectra for each channel are given for comparison. The simulated data in Fig. 7a was generated using the

**Fig. 6 | Four-channel signal decomposition using the gain matrix. a, c** Show the simulated coupled spectra. **b, d** Illustrate the corresponding decomposed spectra. The 'signal' is the primary signal, and the 'interf.' is the interfering coupled signal. The four $^1$H channels and four $^{13}$C channels experienced synchronous pulses. Four samples were used, with each sample assigned one $^1$H and one $^{13}$C nucleus. A hard pulse with a duration of 20 μs was applied to each nucleus, and the average radio frequency amplitude was adjusted to 12.5kHz to achieve a flip angle of 90°.

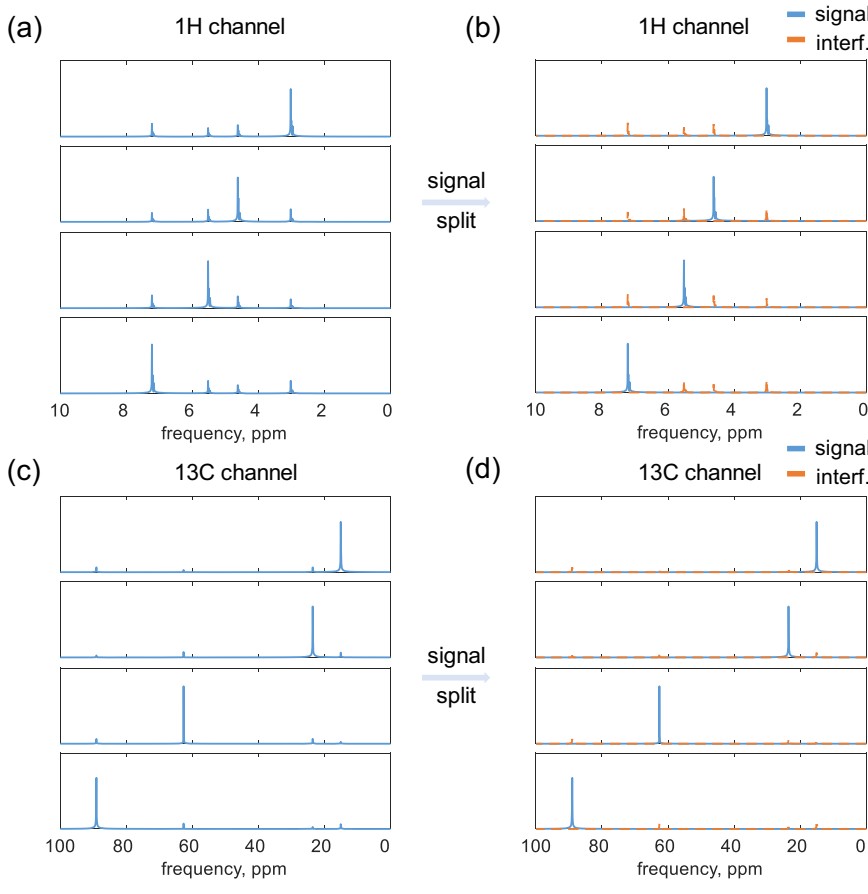

RF chain model. In particular, both the FIDs and noise components of the original signals underwent reception coupling **G**. The coupled signals are fully filtered, indistinguishable from the noise. Figure 7b gives the decomposition results for experimental data obtained with 2 water samples, the primary peaks, i.e., 4 ppm and –1.8 ppm were distanced by applying a shimming gradient. Compared to the original spectrum (gray lines), the split one (colored lines) removed over 90% of the coupled components, while maintaining the individual signals. Note that the BSS for instantaneous mixtures returns an estimated demixing matrix, the recovered signal is given by multiplying this matrix with the coupled signal, so the lineshape of the spectrum is preserved. When handling the highly parallel probes, the spectral differences between signals from two channels may appear minor, especially amidst the complexities of spin systems, both SNR and linewidth can suffer due to decreased shimming quality. To investigate these factors, we present the figure of merit results for the SOBI method in the context of a large number of mutually uncorrelated sources. Note that when dealing with partially correlated source signals, the ability to separate them diminishes rapidly as the number of sources increases, as discussed in literature[62]. Each channel has a single harmonic wave described by $s_m = \exp((i2\pi f_m - \alpha)t)$, where $f_m = m$ and $\alpha$ is the decay rate, the corresponding linewidth in Hz is $\alpha/\pi$. Each source signal involves 5000 sampling points over 10 seconds, with a fixed SNR of 30 dB. The coupling matrix is randomly generated as an $N$ by $N$ complex matrix for $N$ channels. The estimation error is calculated based on the mixing matrix criterion[62].

$$C_A = \frac{1}{N} ||\mathbf{E} - \hat{\mathbf{A}}^{-1}\mathbf{A}||_1 \qquad (5)$$

where **E** is the identity matrix, $\hat{\mathbf{A}}$ is the estimated coupling matrix in which the permutation and scale indeterminacies are removed, and **A** is the real coupling matrix. The calculated $C_A$ is shown in Fig. 8. Performance

improves with more sources, indicating smaller estimation errors and robust separation for uncorrelated sources. Larger linewidth $\alpha$ results in larger $C_A$, for example, with the linewidth decreasing from $0.4/\pi$ to $0.2/\pi$, the estimation efficiency gets an improvement factor of 2 (3 dB). It implies the potential application in dense arrays with enhanced shimming quality. Furthermore, Supplementary Note 8 offers insight into spectral resolution at low SNR signals, indicating that two signals with a 3 Hz spectral difference can be discerned at SNR=30 dB, with a finer resolution of 1.5 Hz achievable at SNR=40 dB.

**Pulse compensation experiment**

The pulse compensation scheme was verified with two channels. Figure 9a illustrates the workflow for the pulse compensation scheme. Initially, the simulated magnetic field was imported into the optimal control module implemented in Spinach to generate the optimal control pulse. Subsequently, a calibration experiment was performed to generate the coupled time domain FID signals, which were then inputted into the BSS module to estimate the coupling matrix $\hat{\mathbf{F}}$, an $n$ by $n$ matrix for an $n$-channel detector. Finally, the cooperative pulse was computed using the equation $\mathbf{p} = \mathbf{F}^{-1}\mathbf{F}_{\text{diag}}\mathbf{p_0}$, where the vector $\mathbf{p_0}$ represents the multiple optimal control pulses, each of which is individually optimized.

Utilizing this workflow, we conducted the $^1$H NMR calibration experiment using two water samples, which were excited with hard pulses. To increase the separation between the two water peaks for easier separation, the global y-shim value was set to a large value of 150,000 on TopSpin, resulting in a spectral difference of 5.3 ppm, for example, water peaks are at –0.7 ppm and 4.6 ppm for detectors 1 and 2, respectively. Parallel FIDs were then acquired for signal decomposition and coupling matrix identification. In Fig. 9b, the coupled component in channel 1 was successfully removed, while the one in channel 2 at -0.7 ppm persisted, indicating a coupling estimation error. This error arises from two factors: Firstly, the shimming

**Fig. 7 | Two-channel signal decomposition pertaining to ¹H spectra using the second-order blind identification method. a** Decomposition of simulated data. The colored lines depict the separated spectra, while the gray lines represent the original spectra. The average signal-to-noise ratio was set to 35 dB. **b** Decomposition of experimental ¹H spectra. Two water samples were simultaneously excited and detected using two striplines. The two water peaks were manually separated by applying a linear gradient. The coupling strength extracted from signal decomposition is 13%. With 64 scans, the average signal-to-noise ratio of the two primary peaks improved to 45 dB. Phase correction and baseline correction were performed after signal decomposition. The two channels are indicated as chn. 1 and chn. 2.

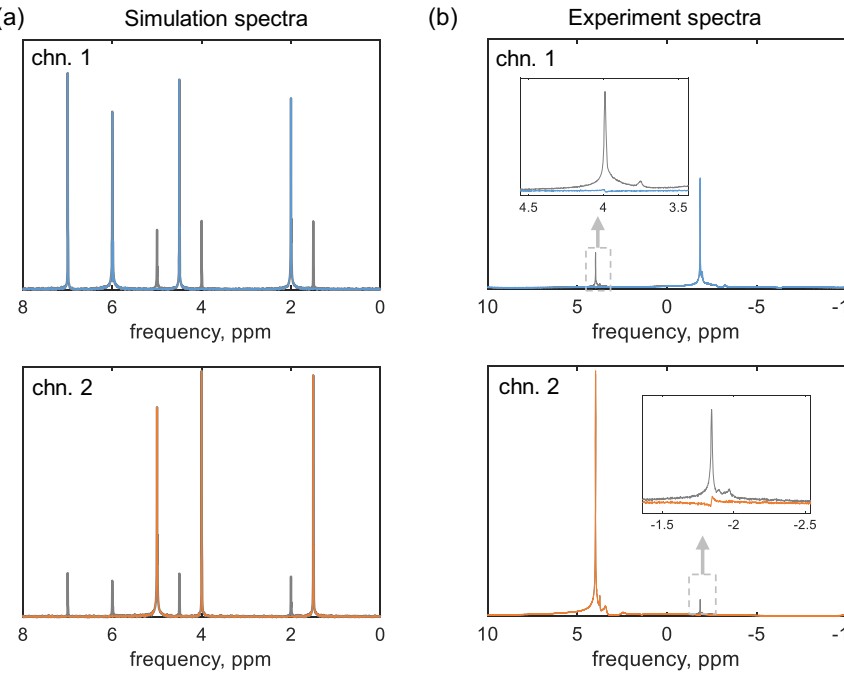

quality in channel 1, the peak exhibits a linewidth of 28 Hz along with a prolonged shoulder on its upfield. Secondly, it stems from suppressed coupling due to an impedance mismatch in channel 2, the coupling from channel 1 to channel 2 is approximately 1%. We subsequently calculated the cooperative pulse following the workflow in Fig. 9a, and tested their efficiency under parallel excitation to verify the pulse compensation scheme. Figure 9c presents ¹H spectra of the two water samples, with the top and bottom plots representing channels 1 and 2, respectively. Gray lines depict spectra obtained with optimal control pulse in single-channel excitation, i.e., $\mathbf{p_0}$ - no couple, for reference. To compare excitation distortions, the three lines in each figure share the same phase correction parameters, optimized based on the lineshape of the reference line. Notably, the shimming was performed using global and local shimming sets, causing a 0.9 ppm shift between the water peaks. In channel 1, spectra for $\mathbf{p_0}$ - coupled show varying amplitude and phase changes due to coupling-induced distortion, while $\mathbf{p}$—coupled yields similar signal characteristics to the reference, indicating successful restoration by the cooperative pulse. However, in channel 2, the lines nearly coincide, suggesting weaker coupling, the strength of 1% was observed in Fig. 9b. Here, the error in coupling estimation led to ineffective compensation by the cooperative pulse. Figure 9d displays spectra of L-alanine and L-valine samples excited with the same optimal control pulses, giving a similar distortion as in Fig. 9c. The closely spaced peaks from L-alanine and L-valine are invisible considering the shimming quality, the peaks in the downfield arise from residual water in $D_2O$ solvent and the peaks at 0 ppm represent TSP.

**Performance of the digital twin**

**Wall clock time.** The wall clock time for the parallel NMR simulation is provided in Table 1. These simulations were carried out on a PC equipped with an AMD Ryzen 5950X 16-core processor, running at a base frequency of 3.40 GHz, and equipped with 64 GB of RAM. Each channel's sample was divided into 72 voxels. In the spin dynamics simulation within each channel, the calculation considered a single ¹H spin, subjected to a hard pulse and acquisition experiment at a 500 MHz magnet. As the number of channels increases, the RF simulation needs to sweep each lumped port to calculate the full $n$ by $n$ scattering matrix, which contributes to the most time-consuming part. The wall clock time of optimal control was given in Supplementary Note 9. For the wall clock

time of large-scale spin system simulation, one may refer to the Spinach literature[63].

**Robustness to poor SNR.** Using the test flow in Fig. 10a, the robustness of signal decomposition and pulse compensation with poor SNR signals were examined through a 2-channel simulation. The EM simulation incorporates a 2-solenoid array with a distance of 1.25 mm. The $B_0$ and $B_1$ fields were processed and imported into Spinach. The spin dynamics simulation involved a ¹H excitation and acquisition experiment, with the spin system for each channel randomly generated, including varying numbers of spins and their chemical shifts.

In Fig. 10b, where the average SNR is 10 dB and minimal $\Delta f = 0.02$ ppm, the cooperative pulse's fidelity exhibits mean values of 0.988 for channel-1 and 0.987 for channel-2, achieving a recovery of 99.3% fidelity from the optimal control pulse. Figure 10c showcases the robustness of $C_A$ concerning SNR, with $C_A$ converging to –17 dB (2%) as SNR increases to 50 dB. Notably, the source signals are weakly correlated, given the presence of cross-coupling between coils and samples during the reception, which causes an additional error. The 2% decomposition error means removing over 95% of coupled components and achieving a similar fidelity level for the cooperative pulse compared to the optimal control pulse, as shown in Fig. 10d.

**Discussion**

We built a digital twin to model parallel NMR experiments, creating a digital environment from which pulse sequence optimization under parallel measurement can be performed. Starting from electromagnetic simulation, the calculated $B_0$ and $B_1$ fields were imported to the Spinach package for spin dynamics simulation, and the FIDs were extracted for signal chain calculation. Based on the model, we adapted cooperative pulses to cancel the inter-coil coupling effects in the excitation stage. In this scheme, pulse optimization only executes on single channels individually, followed by a forward pulse compensation, which can be a potential approach for parallel pulse sequence design. The optimization procedure avoids including inter-channel coupling and allocates its degree of freedom to address specific samples, making this scheme efficient for deployment in highly parallel detectors, in which the $B_1$ field is confined to the coils. In the future, one may also design NMR coil arrays in which the field-level combination becomes notable. In such scenarios, the pulse compensation based on the coupling

matrix may not suffice. Exploring parallel transmit design, as extensively addressed in MRI[20,22,64], could be beneficial.

The cable connections between the probe and spectrometer exert a substantial impact on coupling strength, a factor not incorporated into the simulation model. Fortunately, parallel excitation and reception share the same RF chain and accordingly, the same coupling matrix, i.e., $G_{norm} \approx F_{norm}$. We applied the BSS method to split the parallel signals and identify the coupling matrix. It is worth conducting a calibration experiment using samples with strong and distinct spectra to characterize the parallel system, and using this information for the general samples.

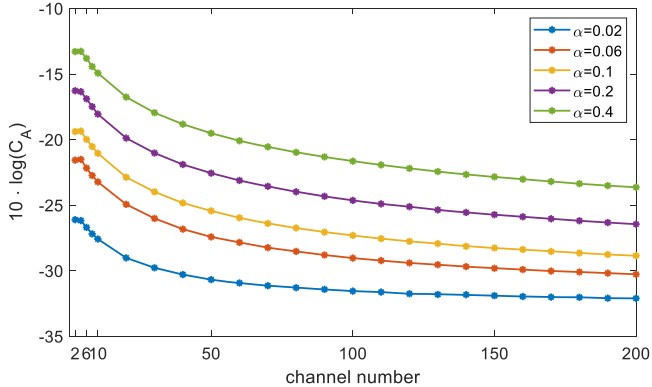

**Fig. 8 | Mixing matrix criterion ($C_A$) as the channel number increases.** The calculation for each channel number was repeated 30 times for which the mean $C_A$ value is displayed. The linewidth of the peak was $\alpha/\pi$.

Since the SOBI algorithm relies on distinct spectral features from each source, the splitting efficiency drastically decreases when the coupled components are located on the shoulders of the primary peaks, as indicated previously, coupling estimation errors escalate with increasing linewidth. Therefore, there is a demand for more robust post-processing methods to facilitate general signal decomposition. These methods should be capable of managing source signals with dense peaks and broad shoulders, as well as naturally correlated source signals. The concepts developed for DOSY may offer avenues in this regard[65–67]. In heteronuclear experiments, such as HSQC, separate pulse compensation for each nucleus is anticipated, due to the frequency-dependent coupling.

This work focused on addressing RF coupling in parallel NMR. In many routine experiments, pulsed field gradients are crucial for the selection of coherence transfer pathways. To generate the required gradient field at each detection site, local gradient coils are utilized in parallel NMR probes. However, this approach can introduce gradient spillover or coupling. Therefore, further research is needed to investigate and address gradient coupling.

## Methods
### Electromagnetic simulations
The simulation scenario includes a solenoid array with a T&M network. The EM simulation was conducted with COMSOL MultiPhysics 6.1 and cross-checked with CST Studio Suite 2022, considering the coil array with water samples inserted. S-parameters were extracted from COMSOL to Matlab R2023a to design a virtual T&M network and build the RF chain model. Spin dynamics calculations and single-channel optimal control were implemented with the Spinach package v2.8[40], the details were provided in Supplementary Notes 10 and 11. For easy reading, the physical variables

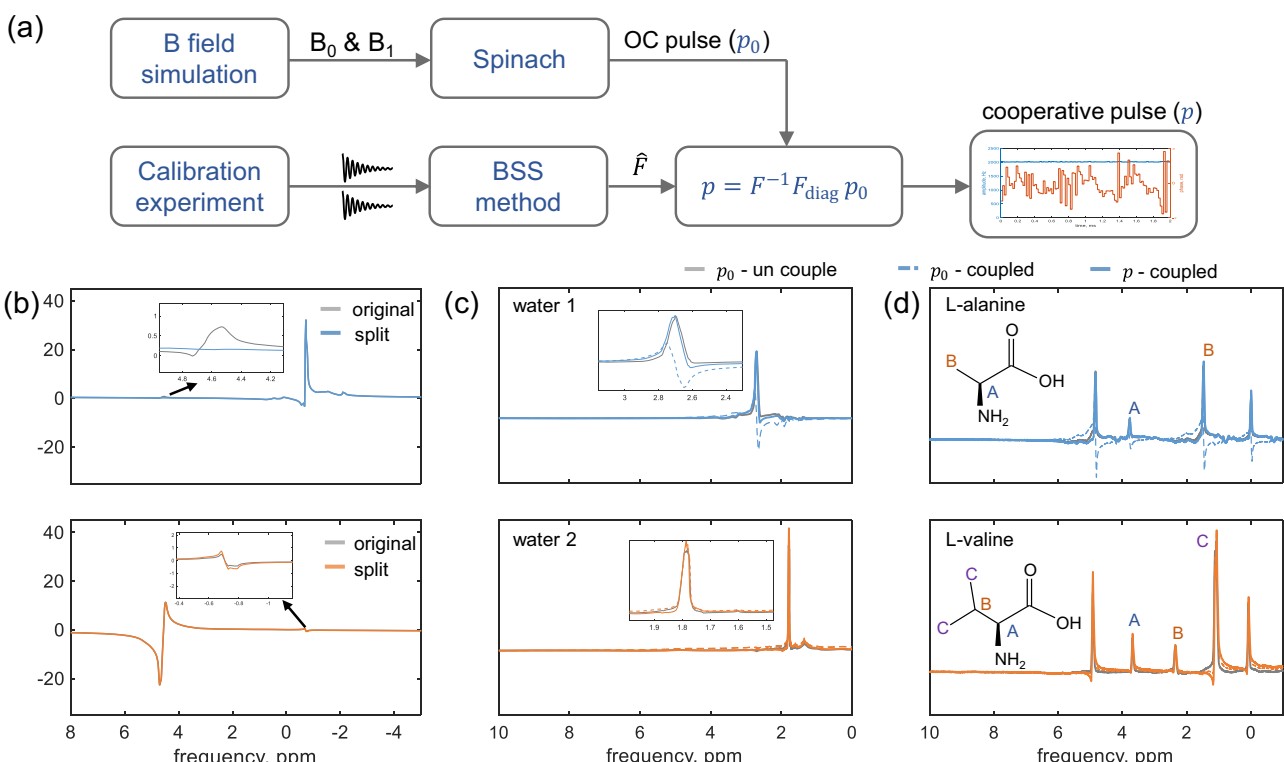

**Fig. 9 | Two-channel pulse compensation experiment. a** Workflow of the pulse compensation scheme. The $B_0$ is the static magnetic field, $B_1$ is the radio frequency magnetic field, OC denotes optimal control, BSS denotes blind source separation, and $F_{diag}$ is the diagonal of coupling matrix $F$. **b** Signal decomposition and coupling identification with 2 water samples, the estimated $\hat{F}_{norm}$ = [0.9998 + 0.0004i, 0.0103 + 0.0201i; 0.0085 − 0.0081i, 0.9998 + 0.0004i].. **c** Two-water $^1H$ spectra, the gray lines and colored dashed lines represent the spectra for an optimal control pulse applied in a single-channel ($p_0$-uncouple), and for parallel excitation ($p_0$-coupled), respectively. Colored solid lines depict spectra with cooperative pulses applied during parallel excitation ($p$-coupled). **d** L-alanine and L-valine $^1H$ spectra, the three lines in each subplot correspond to the same excitation as in (**c**). The left-side peaks originate from residual water, and the right-side peaks from Tri-methylsilylpropanoic acid (TSP). The shimming quality renders the closely spaced peaks invisible. The same phase correction parameters were applied to the three lines in each subfigure in (**c**, **d**).

**Table 1 | Wall clock time (seconds) of the parallel NMR simulations**

| channel number | DC simulation | | RF simulation | | Process B field | spin dynamics | Total time |
|---|---|---|---|---|---|---|---|
| | DOF | time | DOF | time | | | |
| 1 | 213497 | 6 | 969449 | 38 | 3.5 | 1.3 | 48.8 |
| 2 | 304195 | 13 | 2172051 | 158 | 21.5 | 1.6 | 194.1 |
| 4 | 587519 | 19 | 2830773 | 433 | 52.0 | 1.8 | 505.7 |
| 8 | 927966 | 24 | 5501691 | 2202 | 377.8 | 3.3 | 2607.1 |
| 16 | 2149472 | 62 | 7771127 | 10523 | 2151 | 10.4 | 12746 (3.5 hours) |

DOF denotes number of degrees-of-freedom to be solved in a finite element simulation. DC denotes a direct current, and RF denotes the radio frequency wave.

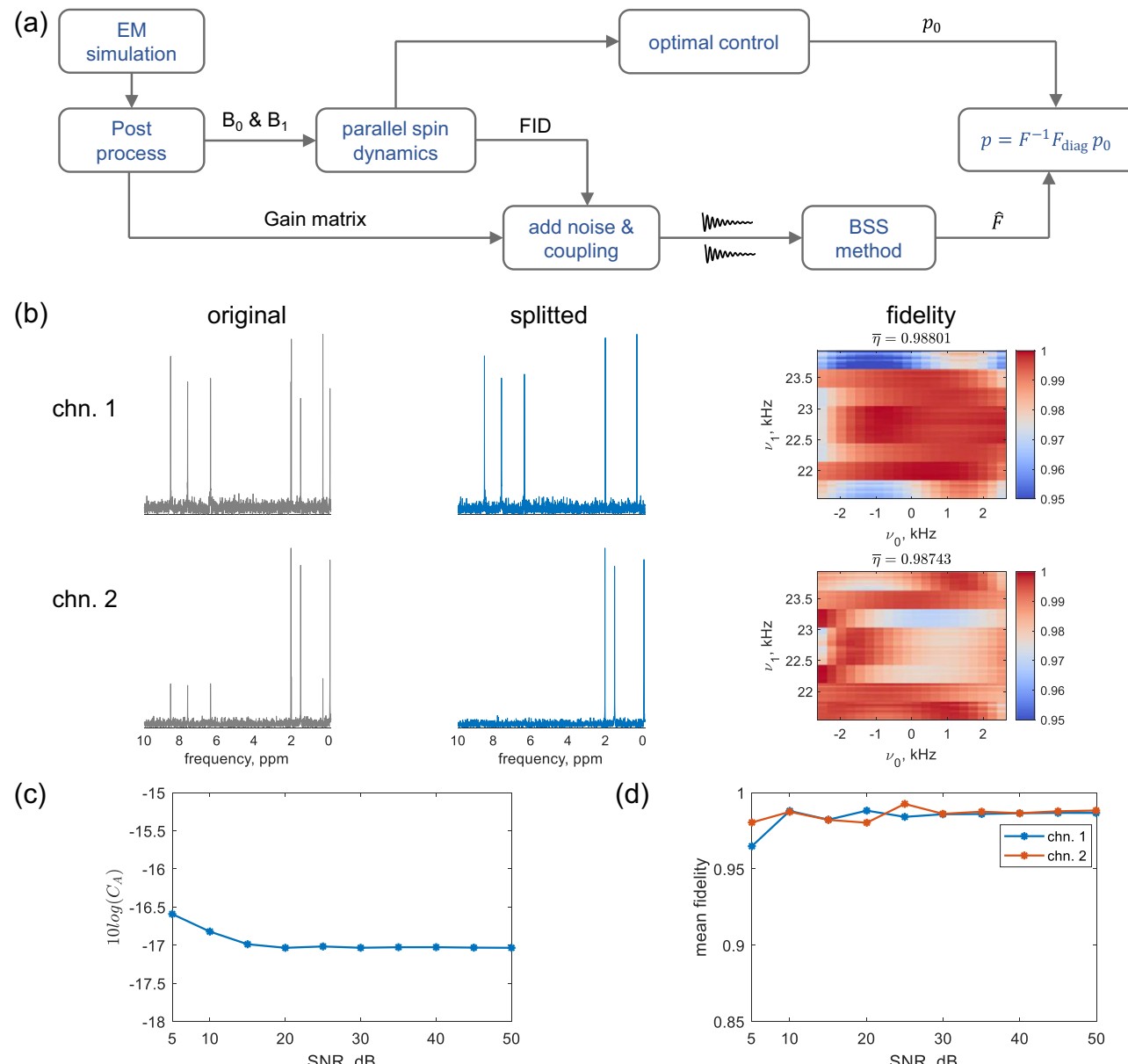

**Fig. 10 | Robustness of the digital twin to poor signal-to-noise ratio. a** End-to-end test flow. EM denotes electromagnetic, $B_0$ is the static magnetic field, $B_1$ is the radio frequency magnetic field, FID denotes the free induction decay, BSS denotes blind source separation, $p_0$ is the optimal control pulse, $p$ is the cooperative pulse, and $F_{\text{diag}}$ is the diagonal of the coupling matrix $F$. **b** Example results for running the digital twin with 2 channels, with a signal-to-noise ratio of 10 dB, and a minimal $\Delta f = 0.02$ ppm (10 Hz). The left, middle, and right panels show the original $^1$H spectrum, the decomposed $^1$H spectrum, and the fidelity ($\eta$) of the cooperative pulse, respectively. $v_0$ is the resonance offset, $v_1$ is the radio frequency amplitude, and $\overline{\eta}$ is the averaged fidelity. **c** Mixing matrix criterion ($C_A$) as a function of the signal-to-noise ratio. **d** Mean fidelity of the cooperative pulse as a function of the signal-to-noise ratio. The mean fidelities of optimal control pulse are $\overline{\eta_1} = 0.9952$ for channel 1 and $\overline{\eta_2} = 0.9951$ for channel 2. The two channels are indicated as chn. 1 and chn. 2 in (**b**) and (**d**).

concerning the EM simulation and spin dynamics calculations were concluded in Supplementary Note 12.

## Parallel signal acquisition experiments

The parallel signal acquisition experiments (Fig. 7b) was carried out using a 15.2 T Bruker NMR system and a custom-built two-stripline probe[68]. Each RF coil was deployed with local shim coils powered by an external DC source to improve $B_0$ homogeneity, an x-gradient was applied to separate the two channels via the global shim (see Supplementary Fig. S17). For synchronous single pulse $^1$H acquisition, the spectra consisted of 64 scans, and 13157 data points were collected for each scan with 20 ppm spectral width.

## Parallel pulse compensation experiments

The 2-channel pulse compensation was conducted using an 11.74 T Bruker NMR system equipped with a 4-stripline probe provided by Voxalytic GmbH, with only channels 1 and 2 utilized. The 2 channels were individually tuned to 500 MHz with a tune and match (T&M) box equipped for the probe, the T&M results were given in Supplementary Fig. S18. The local shimming was powered by a 28-channel shim current source[68]. Pulse calibration was completed within the single channel nutation experiment. Specifically, the nutation experiments were conducted on channel-1 and channel-2 separately to determine the RF amplitude at the suggested maximum power level, 20 W. Then the power level was scaled down to deliver an RF amplitude of 30 kHz, i.e., 8.3 $\mu$s of 90 degree pulse, supposing that $B_1$ amplitude is proportional to $\sqrt{P}$ when the power level located in the linear range of the amplifier. The optimal control pulse has a nominal RF amplitude of 30 kHz, lasts 8 ms, and comprises 4000 time bins. Two channels share the same acquisition parameters, including 3072 data points, a relaxation delay of 5 seconds for water and 10 seconds for L-alanine and L-valine, and a sweep width of 20 ppm for $^1$H acquisition. A total of 64 scans were recorded, with a receiver gain set to 10 for water and 18 for L-alanine and L-valine.

## Sample preparation

The distilled water was prepared for the calibration experiment (Fig. 9b). A solution of 250 mM L-valine and L-alanine (Sigma-Aldrich) samples were prepared in $D_2O$ (99.9%, Sigma-Aldrich) with 10 mM 3-(Trimethylsilyl) propionic-2,2,3,3-d4 acid sodium salt (TSP) as reference compound for verifying the pulse compensation. The chemical shifts of L-alanine and L-valine are given in Supplementary Note 14. The samples were filled into the syringes and manually pumped into the individual fluidic chamber of the dedicated detector.

## Data availability

The data supporting the findings of this work are available from the corresponding authors upon reasonable request.

## Code availability

The Multiphysics simulation code is accessible in the repository https://github.com/kikioh/digital-parallel-NMR.

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

## Acknowledgements

We are supported by the Joint Lab Virtual Materials Design (JLVMD) of the Helmholtz Association, Germany. J.G.K. acknowledges support from the ERC-SyG (HiSCORE, 951459). Y.C., J.G.K., and M.J. were supported by the DFG under contract AN 984/10-1. N.M., M.J., B.L., and J.G.K. acknowledge

partial support from CRC 1527 HyPERiON. The authors acknowledge the support of the Helmholtz Society, both through the program Materials Systems Engineering, as well as for financing the 15.2 T magnet system. We appreciate the suggestions on Spinach simulations provided by Prof. Ilya Kuprov.

## Author contributions

N.M. and J.G.K. conceived of the initial digital twin. M.H. performed the simulations with input from all coauthors. Y.C. and M.J. designed and fabricated the 2-stripline probe. M.H., D.F., N.M., Y.C., and D.B. conducted the experiments. M.H., D.F., N.M., and Y.C. drafted the manuscript. All authors reviewed and refined the manuscript. J.G.K., B.L, and N.M. provided supervision and secured the funding.

## Funding

## Competing interests

J.G.K. is a shareholder of Voxalytic GmbH, a spinoff company that produces and markets microscale NMR devices. The other authors declare no competing interests.
