## [Peer Review File · Communications Engineering]

Reviewers' comments:

Reviewer #1 (Remarks to the Author):

The paper describes a "digital twin" approach to decouple the transmit and receive paths for a parallel NMR coil array. Overall, the concept is interesting and the paper is well-written. Here are some points for the authors to consider:

1. In general, the B1 field distribution depends on sample properties (conductivity, permittivity, and permeability, which can vary with position). How would non-uniform sample properties be handled within this simulation framework?
2. The use of the magnetostatic approximation should be more carefully justified. At 500 MHz, the electromagnetic wavelength inside water is only 6.7 cm, so phase shifts due to wave propagation may become significant.
3. Line 161: "In Spinach, when defining the spin system, interactions between different samples are neglected
162 under the assumption of no interaction between samples." -> This assumption needs to be more carefully justified. For example, it is possible for coupling with nearby coils to generate radiation damping effects on the spin dynamics.
4. Section 3: The distinction between "circuit-level" and "field-level" coupling is hard to follow. All inductive coupling in the system can be described by a mutual inductance matrix if the magnetostatic approximation is valid. What, then, is "field-level" coupling?
5. Are coil losses (e.g., finite Q) included in the simulation model? Please discuss.
6. Figure 6: The range of chemical shifts shown in these spectra is not realistic, particularly for the 1H channel. Real-world 1H spectra have a much narrower range of chemical shifts.

Reviewer #2 (Remarks to the Author):

The manuscript “A digital twin for parallel NMR spectroscopy” presents an approach to parallel NMR.

Parallel encoding/reconstruction received great attention in MRI and the authors reproduce many steps already considered within MRI, but I found little to no trace of such previous works in their manuscript. I suggest the author to consider their approach in the light of K. Pruessmann NMR Biomed. 2006; 19: 288–299 for a general description of a parallel excitation/detection system. The coupling is described by a linear system which can always be diagonalized to disentangling the signal from each receiver or control independently each transmitter once the scattering matrix is known.

Within the receive phase the coils’ coupling is responsible for coupled noise among channels and a general SNR reduction. This is one of the reasons why parallel reception is preferably performed with decoupled coils. In parallel MRI it can be difficult to decouple multiple resonators especially for the transmit phase when preamplifier decoupling is not feasible. For this reason, active decoupling for transmit phase found more interest in the literature and examples can be found in Yang X. et al, US Patent 7,336,074 (2008), numerical simulations were performed in Vernickel P. et al, MAGMA 2006;19S:19 and Galante A. et al Proc. Intl. Soc. Mag. Reson. Med. (ISMRM 2017 conference), p. 5608. In this respect the idea behind the manuscript is not new but this is not clearly readable in the text.

I also found obscure some steps in the authors reasoning. They consider (line 128, “Since unloaded coupling is typically weak, often smaller than -40 dB, each coil could be individually tuned and matched based on the diagonal elements of S_c .”) a low level of coupling among the non-resonant coils without any clear quantification of the applicability limit of their approach. When we add the tuning elements the coupling increases (see fig. S2c) and I do not understand how each channel can be individually tuned and matched. Indeed, a set of coupled resonating elements generates several resonant collective modes and splitting of resonant frequencies. This is a main practical issue that makes unmanageable the experimental tuning/matching procedure for more than 2-3 coupled channels. It is not clear to me how the authors could solve this problem.

I also do not understand the asymmetric behaviour of channels 1 and 2 in the two coils simulation (Figure 4). The explanation provided at lines 229-231 sounds obscure: the physical system is symmetric under the channels labels swap while the results aren’t.

The statement about the transfer efficiency (line 265) is counterintuitive: when coupling is present the power sent to a channel transfers to other channels and is dissipated by the amplifier load thus reducing efficiency.

About the SOBI algorithm I have some doubts about the way the noise was added (line 310, “The simulated signals in Fig. 7(a) were produced by parallel modeling, and noise was added manually before

performing the decomposition.”) since the noise in different channels has some degree of correlation due to coupling.

The Supplementary file is rich in information, but some points remain to be clarified. At line 18 is stated again that “coupling is sufficiently weak that each coil is tuned and matched individually”. This qualitative statement is compatible with considering all S_{11} , S_{12} , S_{21} , S_{22} matrices as diagonal matrices and the symmetry of the coil distribution suggests that they are also proportional to the identity matrix. Under this assumption G and F matrix should be diagonal as well. If the mathematical treatment is exact it should work in the non-diagonal case as well whatever the ratio between diagonal and non-diagonal element is. If the mathematical treatment is approximate it is unclear which is the approximation.

At line 32 the source voltage wave br is not clearly defined.

It is my opinion that the manuscript requires some more clarifications. Its novelty content is reduced if we consider the existing literature and is unclear the admissible coupling among coils that makes feasible the experimental implementation of the procedure. After some improvements I suggest it could better fit a lower impact journal.

Reviewer #3 (Remarks to the Author):

The authors of this article present a framework for parallel NMR on both parallel pulse sequence optimization and parallel signal decomposition. The framework for parallel pulse sequence design is based on full wave simulations of magnetic fields in commercial software, fed to an open-source spin-dynamics simulation also applying the pulse optimization by optimal control.

The purpose is to enable the increase of sample throughput i.e. by detecting multiple samples in parallel. The coupling between these coil elements is handled through electromagnetic simulations and then optimal control to exploit the coupling information to compensate for coupling effects. That these actions primarily happen in computer software has motivated the catchy title of a digital twin for parallel NMR spectroscopy, presumably.

The experimental proof of principle shows results with a two-channel setup with two water samples. Numerical results of other spin systems are shown for other parts of the workflow.

It is a well-written paper, with a potentially high impact for the field of NMR. Albeit it is a very interesting new perspective, I recognize a few issues related to the claims put forward.

Since it is a digital twin, I miss more rigorous testing of the method, with more than just two or eight channels, and from end to end. It is fair the experimental setup contains only two channels. Experimentally, I do however miss a more challenging spin system than two water samples.

It is mentioned how the parallel NMR is related to a similar concept in MRI. Parallel signal reception in MRI is standard with 32 channels or more. There the higher throughput manifests as accelerated acquisitions of volumes rendering the scan time very short compared to when MRI was introduced decades ago. For two decades parallel RF transmission has also been developed and it is mentioned that parallel transmission is indeed securing better excitation abilities. Siemens now sells clinical workhorse 3T MRI systems with 2 Tx channels, and 7T systems with 8Tx as an option in research mode. Research systems and third-party vendors however develop more than 24 Tx channels, ad hoc. Hence, there exists a huge, increasing body of literature in MRI on the topic of multichannel receive and transmit. One issue with the paper is that it does not sufficiently review important aspects from there. In particular, because the field of MRI has struggled and solved a lot of issues related to the author's work, essentially to gain throughput and gain advantage of the more advanced hardware, and this must be properly referenced. There is also the chance that issues solved here in this paper actually have been solved in the other fields.

For example, it is mentioned how a multi-Tx optimal control is "causing the optimization to be time-consuming and convergence is challenging", and further "As the most time-consuming step in the workflow". The time penalty is obvious, but how severe is it at all, and the convergence claim is not backed by evidence. In the NMR setting, one could argue that a time penalty is not an issue because the samples are patient, and the framework, as presented, does not make any attempt to tailor the pulses to the actual situation in the magnet, rather it exploits COMSOL generated fields. In MRI, the situation is different. Robust universal pulses may be optimized in advance also, but better results may be achieved with patient-tailored multi-Tx optimization exploiting experimental B0 and B1 maps measured on a per-patient basis. There the time-penalty of multi-Tx is considerable. And a rich body of literature has addressed this issue, also with optimal control methods very much related to the LBFGS-GRAPE method implemented in Spinach. Could the authors please emphasize why their method is new and that the previous optimal control methods e.g. used in MRI can not solve the problem of the authors.

Please be more clear about how you solve your problem exploiting the S matrix and how the problem in MRI is solved (not using the S matrix). The key reason of your method could be elaborated.

Please give an account of the workflow time consumption. I didn't find any wall-clock times. Please also be more specific about the settings of the LBFGS. I advise establishing a repository with scripts used to run the method, and data, to increase transparency.

The showcase of one waveform of 100 time bins seems extremely easy and does not seem to justify the claims of the experimental challenges. I would advise the authors to present a more difficult spin system e.g. heteronuclear and a more advanced pulse sequence target. It is stated that the method can handle such a case, but it is not shown.

High throughput is mentioned as a purpose, but this is not justified. It is obvious the method has the potential to do it, but separating two water signals does not guarantee success in other spin systems. Related to the time consumption remark, we do not know at this point whether it would be faster or not to measure the two water samples separately in serial. Even if this is a proof of principle, I'm not convinced about the higher throughput. Hence, by simulations, it would be good to stress test your setup. One could imagine a multi-channel micro coils setup on a chip. How many channels can you separate and with what resolution, spatially and spectrally.

Please discuss the resolution at which your signal decomposition works. It would obviously be field strength dependent, but also dependent on spacial constraints e.g. of the bore size.

The title includes "NMR". In my book "NMR" is both solid-state and liquid-state. Solid-state often includes magic angle spinning. While your method is digitally-based where many practical difficulties are non-existent (as it would be for a parallel MAS setup) I would still imagine that a digital high throughput twin for par-MAS-ssNMR would be challenging. If you agree, please limit your claim to what you have shown: few-channel par-1sNMR. Use your stress testing on number of channels if you desire to expand your claim.

Response to Reviewers

Dec. 1, 2023

Please find our point-by-point responses to the three reviewers. We sincerely appreciate the reviewers for their detailed feedback, which we believe has contributed significantly to enhancing our article. For ease of reading, we have highlighted our responses in blue, and likewise, all modifications to our article are highlighted in blue.

Reviewer 1

The paper describes a "digital twin" approach to decouple the transmit and receive paths for a parallel NMR coil array. Overall, the concept is interesting and the paper is well-written. Here are some points for the authors to consider:

We appreciate the reviewer's positive comment.

1. In general, the B1 field distribution depends on sample properties (conductivity, permittivity, and permeability, which can vary with position. How would non-uniform sample properties be handled within this simulation framework?

In our simulations we assumed a uniform sample in terms of position and magnetic susceptibility. When handling a non-uniform sample, position-dependent properties could be defined in a COMSOL simulation, i.e., $\sigma(r)$, $\epsilon(r)$, $\mu(r)$. One can define such functions (analytical, interpolative, or piecewise analytical, etc.) in the "Model Builder->Component->Definitions", and specify the corresponding function in the "Material Contents" section. A non-uniform permeability has a significant impact on the B_0 and can be similarly specified in a static magnetic field simulation ('mfnc' module).

2. The use of the magnetostatic approximation should be more carefully justified. At 500 MHz, the electromagnetic wavelength inside water is only 6.7 cm, so phase shifts due to wave propagation may become significant.

Thank you for raising this point. Exactly, when the sample size is comparable to the wavelength, it is crucial to consider phase shifts. We employ COMSOL's 'emw' module for B_1 calculations. This simulation solves the electric field wave equation $\nabla \times \mu_r^{-1}(\nabla \times \mathbf{E}) - k_0^2 \left(\epsilon_r - \frac{j\sigma}{\omega\epsilon_0} \right) \mathbf{E} = \mathbf{0}$, which encompasses wave propagation. We present the B_1 phase

results with a modified color bar in Figure S6(c), also shown here. The length of the water sample is set to 2.5 mm (3.7% of the wavelength). Figure R1(b) displays the phase shift inside sample 1. The shifts in the center are small, while large shifts exist in the two outer edges, close to the solenoid ends, and the maximum shift is approximately $2.8\% \times 2\pi$. The B_1 field simulation doesn't assume a magnetostatic approximation but relies on a linear combination principle that the total field is the superposition of the field generated by each coil. This principle should apply to wave propagation cases.

Accordingly, we revised the paragraph on page 5: "The RF simulation was carried out using the COMSOL 'emw' module, employing a frequency domain solver that solved the electric field wave equation through the finite element method."

Under the linear combination principle, We completed post-processing in Matlab, to extract the combination coefficients for each voxel. The time scale of the voxel-by-voxel sweep is seconds, as evidenced by our wall-clock time data in Supplementary Table 3. For a single-channel simulation, the sample is divided into 72 voxels, with a total processing time of 3.5 seconds for both B_0 and B_1 fields. However, the time increases rapidly with an increase in the number of channels.

Figure R1 B_1 phase of 4-solenoid coil array, with values given in radians divided by π . (a) Phase on the xoz surface, with each plot corresponding to an individual coil, with only coil 1 being excited. (b) Phase of 72 voxels for sample 1.

3. Line 161: "In Spinach, when defining the spin system, interactions between different samples are neglected 162 under the assumption of no interaction between samples." This

assumption needs to be more carefully justified. For example, it is possible for coupling with nearby coils to generate radiation damping effects on the spin dynamics.

Thank you for bringing up this issue. We addressed the sample-sample coupling by incorporating circuit-level modeling, with the approximation that the radiation damping field is exactly determined by the FID current. Further details can be found in Supplementary section S10. Finally, we concluded that, for a typical value $B_{\text{rad}} = 50$ Hz, the coupled radiation damping field is approximately equal to 5 Hz, considering 10% coupling strength.

Accordingly, we updated the paragraph on page 7: "In Spinach, when defining the spin system, interactions between different samples are neglected under the assumption that sample-sample coupling, e.g., via radiation damping, can be disregarded, see Supplementary section S10."

4. Section 3: The distinction between "circuit-level" and "field-level" coupling is hard to follow. All inductive coupling in the system can be described by a mutual inductance matrix if the magnetostatic approximation is valid. What, then, is "field-level" coupling? Indeed, a coupling matrix suffices to characterize the lumped coupling. These two descriptions provide insight into the calculation of B_1 , wherein the term 'field-level coupling' is more accurately construed as a linear combination of the B_1 fields generated by each coil. We use two matrices, F and T , to compute B_1 from pulse p . In doing so, we disentangle the mutual coupling into a circuit-level combination (F) and a field-level combination (T). Specifically, F represents that coil currents result from the linear combination of parallel pulses, while T represents the direct field spillover, signifying that B_1 results from the superposition of B_1 generated by each coil. Considering that T is extracted from full-wave simulation and exhibits position dependence, this matrix includes wave propagation, thus working beyond the magnetostatic approximation. To avoid confusion, we have revised the term 'coupling' to 'combination' and clarified the 'circuit-level' and 'field-level' combination (page 8 in the main text).

5. Are coil losses (e.g., finite Q) included in the simulation model? Please discuss.

We recognize that finite Q leads to energy loss, affects pulse transient response, and influences the radiation-damping field during reception. Concerning energy loss, in the RF chain modeling (refer to Equation S5), the power loss in the coil is given by $P = \text{real}(V_c \cdot$

I_c^*), while the power delivered to B_1 is given by $Q = \text{imag}(V_c \cdot I_c^*)$. For the case of a high-Q coil, its terminal voltage and current should closely approximate orthogonality. The coil resistance is also employed in calculating the noise voltage, as outlined in Equation S12. Regarding the transient response, we have neglected it, the RF simulation conducted in the frequency domain yields steady-state results. This issue was discussed in the literature, concerning simulating the transient response[1] and measuring the impulse response function[2], and we cited these papers in the EM simulation part (page 6 in the main text). The radiation damping was disregarded in the FID calculation, and we discussed the strength of coil-coupled induced radiation damping in Supplementary section S10.

6. Figure 6: The range of chemical shifts shown in these spectra is not realistic, particularly for the ^1H channel. Real-world ^1H spectra have a much narrower range of chemical shifts.

Thanks for pointing out this issue. We redefined the chemical shifts in the Spinach simulation and modified Figure 6. Here the ^1H bandwidth is set to be 10 ppm and the ^{13}C bandwidth is set to be 100 ppm.

Reviewer 2

The manuscript “A digital twin for parallel NMR spectroscopy” presents an approach to parallel NMR.

Parallel encoding/reconstruction received great attention in MRI and the authors reproduce many steps already considered within MRI, but I found little to no trace of such previous works in their manuscript. I suggest the author to consider their approach in the light of K. Pruessmann NMR Biomed. 2006; 19: 288–299 for a general description of a parallel excitation/detection system. The coupling is described by a linear system which can always be diagonalized to disentangling the signal from each receiver or control independently each transmitter once the scattering matrix is known. Within the receive phase the coils’ coupling is responsible for coupled noise among channels and a general SNR reduction. This is one of the reasons why parallel reception is preferably performed with decoupled coils. In parallel MRI it can be difficult to decouple multiple resonators especially for the transmit phase when preamplifier decoupling is not feasible. For this

reason, active decoupling for transmit phase found more interest in the literature and examples can be found in Yang X. et al, US Patent 7,336,074 (2008), numerical simulations were performed in Vernickel P. et al, MAGMA 2006;19S:19 and Galante A. et al Proc. Intl. Soc. Mag. Reson. Med. (ISMRM 2017 conference), p. 5608. In this respect the idea behind the manuscript is not new but this is not clearly readable in the text.

We appreciate the valuable comments and highly related references provided by the reviewer. We agree that our pulse compensation method is one kind of active decoupling, similar to the decoupling method in MRI, as the ISMRM papers proposed. We have carefully reviewed these papers and herein clarify the uniqueness of our work, detailing how it can enhance parallel NMR experiments. In the routing of the parallel NMR experiments, one needs to connect the parallel probe/detector to the spectrometer and perform tuning and matching. The coupling matrix changes with the sample loading and the TM process. It's therefore hard to determine the coupling only using simulation. In Yang X. et al's US Patent, the coupling measurement relies on a port sweep procedure, wherein each element in the coupling matrix requires one measurement. For comparison, our method is based on the signal decomposition of the detected parallel signals, which should be completed in one measurement, i.e., parallel acquisition. Also, we employed active decoupling to apply the optimal control pulse in the parallel case.

Accordingly, we have reviewed the literature regarding MRI parallel transmission and decoupling schemes in the introduction, specifically highlighting the active decoupling papers that inspired our work. We included Figure 8 in the main text to clarify the pulse compensation scheme, where the coupling matrix was estimated from the signal decomposition of the experimental data. This matrix was then combined with a single-channel optimal control pulse to design a cooperative pulse, the excitation efficiency of which was presented in Figure 8(c-d).

I also found obscure some steps in the authors reasoning. They consider (line 128, "Since unloaded coupling is typically weak, often smaller than -40 dB, each coil could be individually tuned and matched based on the diagonal elements of S_c .") a low level of coupling among the non-resonant coils without any clear quantification of the applicability limit of their approach. When we add the tuning elements the coupling increases (see fig. S2c) and I do not understand how each channel can be individually tuned and matched. Indeed, a set of coupled resonating elements generates several resonant collective modes and

splitting of resonant frequencies. This is a main practical issue that makes unmanageable the experimental tuning/matching procedure for more than 2-3 coupled channels. It is not clear to me how the authors could solve this problem.

Thanks for pointing out this issue, we recognized that coupling distorts the matching and quantified the S_{CM} as a function of unload coupling strength, considering a 2-solenoid array, see Fig. R2(a) below. When $S_{C,21} < -60$ dB, the $S_{CM,11} < -20$ dB, which we regard as a good match. If the coupling increases, individual matching becomes unacceptable, so the TM network needs to be optimized. For this purpose, we start with each circuit connected to an isolated TM circuit (Fig. S2(b)), in which the coupling cannot be removed. Then we applied the Nelder-Mead simplex algorithm implemented in the Matlab function 'fminsearch.m' to optimize the capacitance values, with the target of minimizing the total reflection coefficient. Figure S3 gives the optimized results of a 16-solenoid array. However, applying this optimization procedure to experiments with strong coupling remains a challenge for us.

Accordingly, we revised the Supplementary section S2 for the concerned tuning and matching issue.

Figure R2 Coupling ratio sweep of a 2-solenoid array. (a) Matched S-parameters as a function of unload coupling strength. (b) The difference of normalized G and F matrix, i.e., Frobenius norm of $G_{\text{norm}} - F_{\text{norm}}$ as a function of matched coupling strength. When $c.s. < 0.35$, the capacitances were calculated from $S_{C,11}$ and $S_{C,22}$, which are slightly different. When $c.s. > 0.35$, the capacitances were optimized to minimize the reflection coefficients.

I also do not understand the asymmetric behaviour of channels 1 and 2 in the two coils simulation (Figure 4). The explanation provided at lines 229-231 sounds obscure: the physical system is symmetric under the channels labels swap while the results aren't.

In Fig. 4(a), the hard pulse has limited bandwidth, and its transfer efficiency on distinct spins depends on their chemical shifts, which leads to the difference between the two lines. This is because the chemical shifts in channels 1 and 2 are assigned to -2 ppm and 5 ppm, respectively. In Figure 4(b), two optimal control pulses are individually generated, i.e., p_{01} and p_{02} , which are both robust to ν_0 and ν_1 inhomogeneity. The coupling distorted their waveform and generated new pulse shapes, i.e., $I_c = F \cdot p_0$. Even though the two channels have the same coil geometry (B_1 pattern), we cannot predict which channel performs better, but the robustness of coupled pulses tends to become worse as the coupling increases.

The statement about the transfer efficiency (line 265) is counterintuitive: when coupling is present the power sent to a channel transfers to other channels and is dissipated by the amplifier load thus reducing efficiency.

Apologies for the confusion, as shown in Eq.(3), if a cooperative pulse is applied, the resulting signal received by the coils is $I_c = F_{\text{diag}} \cdot p_0$ (right hand side), which means an optimal control pulse, or its waveform, is recovered in each channel. The remaining task is to adjust the power level to the designed value.

To avoid confusion, the statement in page 11 is revised as:

”Under a wide range of coupling strengths, as long as Eq.(3) remains solvable, a cooperative pulse can achieve comparable transfer efficiency relative to the case in which no coupling exists and optimal control pulses apply on isolated channels.”

About the SOBI algorithm I have some doubts about the way the noise was added (line 310, “The simulated signals in Fig. 7(a) were produced by parallel modeling, and noise was added manually before performing the decomposition.”) since the noise in different channels has some degree of correlation due to coupling.

Thanks for pointing out this issue, we recognized the missed noise coupling in calculating the detected signals. Similar to the FIDs, the noise contributes to the open-circuit voltage at the coil terminal and obtains a gain (coupling) when we measure the signals on the matched loads. We reproduced the signals and conducted the signal decomposition for

Fig. 7(a). The relevant paragraph is modified as:

”The *simulated data* in Fig. 7(a) were generated using the RF chain model outlined in Supplementary section S2. In particular, both the FIDs and noise components of the original signals underwent reception coupling G .”

The Supplementary file is rich in information, but some points remain to be clarified. At line 18 is stated again that “coupling is sufficiently weak that each coil is tuned and matched individually”. This qualitative statement is compatible with considering all S_{11} , S_{12} , S_{21} , S_{22} matrices as diagonal matrices and the symmetry of the coil distribution suggests that they are also proportional to the identity matrix. Under this assumption G and F matrix should be diagonal as well. If the mathematical treatment is exact it should work in the non-diagonal case as well whatever the ratio between diagonal and non-diagonal element is. If the mathematical treatment is approximate it is unclear which is the approximation.

Although each coil is tuned and matched individually, we use full S_C for the subsequent RF chain modeling. Here we give an empirical explanation to clarify the approximation, rewriting the F and G matrices,

$$\begin{aligned} F &= \frac{1}{\sqrt{Z_0}}(E - S_C)S_C^{-1} \cdot (E - S_C S_{11})^{-1} \cdot S_C S_{12}, \\ G &= \frac{1}{2}S_{21} \cdot (E - S_C S_{11})^{-1} \cdot (E - S_C). \end{aligned} \tag{1}$$

Ignoring the coefficients $\frac{1}{\sqrt{Z_0}}$ and $\frac{1}{2}$, and considering that $S_{12} = S_{21}$, since a purely capacitive TM network is lossless and reciprocal. The remaining terms in F and G are exactly the same, except for their relative location.

When each coil is individually tuned and matched, S_{11} , S_{12} , S_{21} , S_{22} are all diagonal matrices, with tiny differences among the elements in each matrix since $S_{C,11}$ and $S_{C,22}$ process tiny difference given by simulation. Note that the two diagonal matrices are commutative, so F and G only differ by a constant $\frac{\sqrt{Z_0}}{2}$ if S_C is also diagonal. However, the small off-diagonal elements in S_C cause a tiny error, this is $G \approx \sqrt{Z_0}/2 \cdot F$. In addition, although the S_C is not diagonal, if each coil is tuned and matched with the same capacitance sets, as we did in the capacitance optimization, the S_{12} and S_{21} are exactly proportional to the identity matrix as you pointed out, in this case, the normalized F and G should be exactly equal, shown in Fig. R2(b). The corresponding matrices at $c.s. = 0.45$ are shown below

$$\begin{aligned}
S_C &= \begin{pmatrix} 0.9780 + 0.2002i & -0.0004 + 0.0018i \\ -0.0004 + 0.0018i & 0.9780 + 0.2003i \end{pmatrix}, \\
S_{12} = S_{21} &= \begin{pmatrix} 0.0097 + 0.0703i & 0 \\ 0 & 0.0097 + 0.0703i \end{pmatrix}, \\
F_{\text{norm}} = G_{\text{norm}} &= \begin{pmatrix} 0.6439 - 0.0176i & -0.0048 + 0.2889i \\ -0.0048 + 0.2890i & 0.6466 - 0.0070i \end{pmatrix}.
\end{aligned} \tag{2}$$

At line 32 the source voltage wave b_r is not clearly defined.

Thank you for pointing out this issue. The Supplementary section S2 is reorganized to smooth the derivation process. Note that the RF chain is modeled by S-parameters, to adapt to this paradigm, FID signals V_{1on} need to be transformed to a reverse wave format on the coil port, and this transform is shown in Eq. S9. Here b_R is the FID-induced reverse wave, and V_{1on} is the FID calculated according to the reciprocity principle.

It is my opinion that the manuscript requires some more clarifications. Its novelty content is reduced if we consider the existing literature and is unclear the admissible coupling among coils that makes feasible the experimental implementation of the procedure. After some improvements I suggest it could better fit a lower impact journal.

Regarding admissible coupling, we acknowledge that the efficiency of pulse compensation diminishes as coupling increases, because direct B_1 field spillover can become significant in strong coupling cases, and our compensation only covers the circuit-level combination. Supplementary section S9 provided the fidelity of cooperative pulses under coupling strengths of 0.09, 0.24, and 0.36, demonstrating that cooperative pulses can maintain 96% of the optimal control pulse's efficiency under these coupling strengths, if the coupling matrix is accurately estimated. Moreover, increasing coupling expands the dynamic range of cooperative pulse amplitudes, potentially straining the power amplifier. Limiting the pulse amplitude is one of the solutions. Following the comments from the third reviewer, our experimental setup involved a 2-stripline array with approximately 2% coupling (see Figure 8), and we anticipate that stronger coupling could be accommodated with amplitude-limited optimal control pulses.

Following the reviewer's helpful suggestions and comments, we carefully added simulation

and experimental results to support the claims in the manuscript, particularly the experimental validation for the pulse compensation. We also expanded on topics including RF chain modeling, optimal control settings, signal decomposition, and end-to-end testing of the digital twin. We now believe that these revisions have significantly improved this work. Therefore, we sincerely request your kind reconsideration of this manuscript.

Reviewer 3

The authors of this article present a framework for parallel NMR on both parallel pulse sequence optimization and parallel signal decomposition. The framework for parallel pulse sequence design is based on full wave simulations of magnetic fields in commercial software, fed to an open-source spin-dynamics simulation also applying the pulse optimization by optimal control. The purpose is to enable the increase of sample throughput i.e. by detecting multiple samples in parallel. The coupling between these coil elements is handled through electromagnetic simulations and then optimal control to exploit the coupling information to compensate for coupling effects. That these actions primarily happen in computer software has motivated the catchy title of a digital twin for parallel NMR spectroscopy, presumably.

The experimental proof of principle shows results with a two-channel setup with two water samples. Numerical results of other spin systems are shown for other parts of the workflow.

It is a well-written paper, with a potentially high impact for the field of NMR. Albeit it is a very interesting new perspective, I recognize a few issues related to the claims put forward.

Since it is a digital twin, I miss more rigorous testing of the method, with more than just two or eight channels, and from end to end. It is fair the experimental setup contains only two channels. Experimentally, I do however miss a more challenging spin system than two water samples.

We would like to thank the reviewer for providing us with detailed and valuable comments. Experimentally, we have validated the pulse compensation using two water samples and samples of L-alanine & L-valine. The results, displayed in Figure 8 of the main text, demonstrate that the efficiency of optimal control pulses is maintained in parallel excita-

tion through compensation.

Regarding the simulation, we presented an end-to-end test in Supplementary section S12. Concerning the 2-channel operation, we integrated EM simulation, spin dynamics calculation, signal decomposition, along with pulse compensation, into a single workflow. Using this workflow, we evaluated signal decomposition errors and pulse compensation efficiency under challenging SNR conditions (5-50 dB).

To accommodate more channels and diverse spin systems, considering that decoupling both the transmit stage and reception stage relies on identifying the coupling, we detailed this matter in Supplementary section S11. Please also refer to the following response.

It is mentioned how the parallel NMR is related to a similar concept in MRI. Parallel signal reception in MRI is standard with 32 channels or more. There the higher throughput manifests as accelerated acquisitions of volumes rendering the scan time very short compared to when MRI was introduced decades ago. For two decades parallel RF transmission has also been developed and it is mentioned that parallel transmission is indeed securing better excitation abilities. Siemens now sells clinical workhorse 3T MRI systems with 2 Tx channels, and 7T systems with 8Tx as an option in research mode. Research systems and third-party vendors however develop more than 24 Tx channels, ad hoc. Hence, there exists a huge, increasing body of literature in MRI on the topic of multichannel receive and transmit. One issue with the paper is that it does not sufficiently review important aspects from there. In particular, because the field of MRI has struggled and solved a lot of issues related to the author's work, essentially to gain throughput and gain advantage of the more advanced hardware, and this must be properly referenced. There is also the chance that issues solved here in this paper actually have been solved in the other fields. Thanks for highlighting these important technical advances, we recognize that MRI has developed enormous multiple-channel applications and have now reviewed the literature in the introduction. We reviewed the concept of 'parallel imaging' and 'parallel transmission', and the subsequently developed parallel pulse optimization for addressing field inhomogeneity, shortening pulse duration, and so on. After reviewing the MRI decoupling schemes, we also point out that the pulse compensation scheme in our work is similar to the active decoupling method proposed in two simulation works published at the ISMRM (citations 29-30), and gratefully acknowledge reviewer 2 for bringing these papers to our

attention. To give novelty to our work, we combined these active decoupling methods with an optimal control pulse, and we estimated the coupling matrix from experimental NMR signals, distinguishing our approach from these previous simulation works and a US patent (Yang X. et al, US Patent 7336074 (2008), also thanks to the reviewer 2 for pointing it out) in which a coupling measurement was conducted by port sweeping.

For example, it is mentioned how a multi-Tx optimal control is "causing the optimization to be time-consuming and convergence is challenging", and further "As the most time-consuming step in the workflow". The time penalty is obvious, but how severe is it at all, and the convergence claim is not backed by evidence. In the NMR setting, one could argue that a time penalty is not an issue because the samples are patient, and the framework, as presented, does not make any attempt to tailor the pulses to the actual situation in the magnet, rather it exploits COMSOL generated fields. In MRI, the situation is different. Robust universal pulses may be optimized in advance also, but better results may be achieved with patient-tailored multi-Tx optimization exploiting experimental B0 and B1 maps measured on a per-patient basis. There the time-penalty of multi-Tx is considerable. And a rich body of literature has addressed this issue, also with optimal control methods very much related to the LBFSGS-GRAPE method implemented in Spinach. Could the authors please emphasize why their method is new and that the previous optimal control methods e.g. used in MRI can not solve the problem of the authors.

Apologies for the lack of rigor in the previous claim. We can empirically show that the time complexity, as control steps increase, is not expected to improve beyond $O(N)$, as shown in the wall-clock time results in Supplementary section S13 and literature[3]. Regrettably, we are unable to provide evidence regarding the convergence of a multi-Tx optimal control approach. The previous claim was deleted.

We also wish to clarify the key reasoning behind the pulse compensation method. Similar to an optimized pulse in MRI, the single-channel optimal control pulse in NMR offers great robustness to the resonance offset, B_1 inhomogeneity, and spin-spin coupling, without utilizing the field map data. In the context of parallel NMR, involving multiple and potentially distinct samples, the pulse applied to each channel must prioritize specific sample excitation while guarding against coupled radiation from neighboring channels. Also, the NMR coils, such as a solenoid or stripline, have uniform B_1 fields confined in their center, resulting in only a weak field coupling contribution from one channel to

others. Therefore, we may only need to decouple the optimal control pulses that are targeted on specific channels, in which case the decoupling is implemented by designing compensation with the estimated coupling matrix. With this approach, we avoid field mapping and make pulse optimization for each channel independent, which grants the pulses more flexibility in addressing field inhomogeneity and spin-spin interactions on specific samples.

Accordingly, the paragraphs in the main text have been revised:

On page 8: "Utilizing the obtained B -map database, the concurrent optimization of pulses across multiple channels enhances robustness against load-induced B -field inhomogeneity, thereby improving single-sample excitation in MRI. However, in parallel NMR involving multiple and potentially distinct samples, the pulse applied to each channel must prioritize specific sample excitation while guarding against coupled radiation from neighboring channels. If the inter-channel coupling details are known, the simultaneous optimization can be transformed into channel-specific compensation. This approach may grant the signal-channel pulse greater flexibility in addressing field inhomogeneity and spin-spin interactions."

On page 14: "The optimization procedure avoids including inter-channel coupling and allocates its degree of freedom to address specific samples, making this scheme efficient for deployment in highly parallel detectors, in which the B_1 fields are confined to the coils."

Please be more clear about how you solve your problem exploiting the S matrix and how the problem in MRI is solved (not using the S matrix). The key reason of your method could be elaborated.

Thank you for highlighting this issue. We recognize that the compensation scheme was not clearly demonstrated in the original text. In response, we have included a workflow figure (Figure 8a) in the main text to clarify the pulse compensation method. A coupling matrix has been derived through signal decomposition of the experimental data and is subsequently integrated with the single-channel optimal control pulse to formulate the cooperative pulse. As clarified above, the S-matrix method is designed to decouple multiple channels, allowing a single-channel pulse to concentrate on a specific sample and address field inhomogeneity and spin-spin interactions.

Please give an account of the workflow time consumption. I didn't find any wall-clock times.

We have calculated the wall-clock time of the multi-physics simulation and optimal control. The results are given in Supplementary section S13. As regards the multi-physics simulation, we tested the 1 to 16-channel variants as shown in the following table. Note that as the number of channels increases, the RF simulation needs to sweep each lumped port, to calculate the full n by n S-matrix, which contributes to the most time-consuming part.

channel number	DC simulation		RF simulation		Process B field	spin dynamics	Total time
	DOF	time	DOF	time			
1	213497	6	969449	38	3.5	1.3	48.8
2	304195	13	2172051	158	21.5	1.6	194.1
4	587519	19	2830773	433	52.0	1.8	505.7
8	927966	24	5501691	2202	377.8	3.3	2607.1
16	2149472	62	7771127	10523	2151	10.4	12746 (3.5 h)

Table R1 Wall clock time (seconds) for the parallel NMR simulation.

Please also be more specific about the settings of the LBFGS.

Thank you for pointing out this issue, we have detailed the description regards the LBFGS-GRAPE algorithm in Supplementary section S7.

I advise establishing a repository with scripts used to run the method, and data, to increase transparency.

Thank you for this suggestion, we will make our data openly available through an online repository.

The showcase of one waveform of 100 time bins seems extremely easy and does not seem to justify the claims of the experimental challenges. I would advise the authors to present a more difficult spin system e.g. heteronuclear and a more advanced pulse sequence target. It is stated that the method can handle such a case, but it is not shown.

Thank you for raising the time bins issue. We agree that practical optimal control pulses would process thousands of time bins, which also offer more degrees of freedom for strict

optimization targets. We set 100 time bins in the simulation to pursue a fast calculation, and the duration 20 μ s for each time bin is enough to cover 5 kHz of bandwidth. In the revised version, we optimized the experimental pulse, with a nominal RF amplitude of 30 kHz, a duration of 8ms, and for 4000 time bins. We have indicated these parameters in the methods and Supplementary section S7.

In experiments, apart from two water samples for coupling identification, we used L-alanine and L-valine in the 2-channel 1 H experiment, the pulse compensation results are shown in Figure 8.

We agree that integrating a heteronuclear experiment and a more sophisticated pulse sequence are important next steps. However, we believe this is outside the scope of the current work, and that the parallel excitation experiment can validate our RF pulse compensation scheme. Our future endeavors will entail heteronuclear experiments, incorporating gradient pulse compensation, with parallel gradient coil hardware already available in our group for testing. Consequently, the initial statement has been omitted, and we have appended the following perspective to the discussion: "In heteronuclear experiments, separate pulse compensation for each nucleus is anticipated, due to the frequency-dependent coupling."

High throughput is mentioned as a purpose, but this is not justified. It is obvious the method has the potential to do it, but separating two water signals does not guarantee success in other spin systems. Related to the time consumption remark, we do not know at this point whether it would be faster or not to measure the two water samples separately in serial. Even if this is a proof of principle, I'm not convinced about the higher throughput. Hence, by simulations, it would be good to stress test your setup. One could imagine a multi-channel micro coils setup on a chip.

Regarding the high-throughput concept, one can anticipate an increase in throughput as N independent detectors and N samples are employed. The N = 4 version has been implemented in an eight-coil probehead with 4 receiver channels [4], and the N = 2 version has been realized in a parallel shimming work[5]. To ensure independent operation, multiple channels need to be decoupled. This work focuses on addressing RF coupling in both the transmit and reception stages. It's worth noting that consideration of gradient coupling may be necessary if parallel gradient pulses are applied, though this is beyond

the scope of this work, but is being actively addressed in our group. Additionally, we also face parallel shimming challenges that limit the sensitivity of parallel detectors, as placing multiple samples at the magnet center is not feasible.

When dealing with the separation of complex spin systems, the BSS method relies on high shimming quality (narrow linewidth), especially when two spectral components are closely situated. In cases where coupled components fall on the shoulders of primary components with large linewidths, separation efficiency significantly declines, as shown in Fig. R3. However, achieving better than 25 Hz linewidth while maintaining a symmetrical peak shape for our parallel detector is challenging, thus hindering the separation test for complex experimental signals. Therefore, we showcase the separation of the experimental signal with a gradient applied, the 2 samples being acetone and isopropanol, see Fig. S15.

Accordingly, in the discussion we point out the demand for a more robust separation method for those cases with coupled-signal overlap.

In the simulation, we tested resolution at poor SNR, the separation of 8 channels for complex spin systems, and the separation error as the channel number increases with varying decay rates (linewidth). For further details, please refer to the subsequent response.

How many channels can you separate and with what resolution, spacially and spectrally. Please discuss the resolution at which your signal decomposition works. It would obviously be field strength dependent, but also dependent on spacial constraints e.g. of the bore size.

Exactly, the present separation ability relies on several factors, including the noise level, the linewidth/shimming quality, the spectral difference, the number of channels, and the coherence level of the source signals if they are correlated. To study one factor, we fixed the others, and we separately discussed the estimation error as a function of the number of channels, signal decay rate (linewidth), and spectral resolution at poor SNR. The detailed results were showcased in Supplementary section S11.

As regards the estimation error for a large number of channels, we tested the BSS method on $N = 2$ to 200 channels, considering that the source signals are uncorrelated. As shown in Figure R3, the C_A measures the distance of the estimated coupling matrix with the real one. Performance improves as N increases, indicating smaller estimation errors and

robust separation for uncorrelated sources. However, a larger linewidth α results in larger C_A , indicating the importance of shimming quality for applying this method.

Figure R3 Mixing matrix error as the channel number increases. $C_A = \frac{1}{N} \|E - \hat{A}^{-1}A\|_1$, where E is the identity matrix, \hat{A} is the estimated coupling matrix, and A is the real coupling matrix. Each channel m has a single harmonic wave described by $s_m = \exp((i2\pi m - \alpha)t)$.

The spectral resolution also depends on the SNR. To discuss the resolution, we studied an 8-channel signal decomposition with simulated data, assuming homogeneous B_0 , as shown in Fig. R4. Each channel has one frequency component, with a minimal frequency difference of 0.006 ppm (3 Hz @ 500MHz magnet) and a local (peak) SNR = 30 dB. The decomposition fails when spectral difference shrinks while maintaining SNR. We further give the resolution as the SNR increases in Table R2.

For those signals detected from complex spin systems, each channel contains multiple frequency components, which gives more abundant statistical information for coupling identification as the SOBI method relies on second-order statistics, in which case the resolution limits may break down. To give evidence, Figure R5 considers an extreme situation, the two peaks from channel-2 and channel-4 have only a 0.001 ppm difference with SNR=30 dB. The signals were successfully split if we check the chemical shifts assigned for each channel.

Moreover, the decomposition is field strength dependent as a higher field corresponds to

a larger frequency difference and improved SNR, both of which make the separation more effective. As regards the spatial constraints, deploying more detectors within a fixed bore increases inter-channel coupling, so that the direct B_1 field spillover can become significant, which leads to correlated sources. While the separation of correlated sources is of interest, it is beyond the scope of this work. As we explore the parallelism limits within a fixed bore, in the future we anticipate a more comprehensive solution regarding highly dense detectors.

Figure R4 Signal decomposition of an 8-channel system using simulated data. Each channel is assigned with one spin, i.e., one chemical shift. The original signal is represented in grey, the split signal is represented in blue. Two signals with a minimal difference of 0.006 ppm (3 Hz @500MHz) and SNR = 30 dB are separated, as illustrated in the zoomed-in view at the top.

SNR, dB	20	23	25	30	35	40	45
resolution, Hz	25	17.5	3.5	3.0	2.5	1.5	1.0

Table R2 Decomposition resolution as the SNR increases, under homogeneous B_0 .

Figure R5 Signal decomposition of an 8-channel system using simulated data. Each channel is assigned a random number of spins, and random chemical shifts in the range [0,10] ppm. The original signal is represented in grey, and the split signal is represented in blue. Two peaks in channel-2 and channel-4 have only 0.001 ppm spectral difference and SNR = 30 dB, see the zoomed-in view.

The title includes "NMR". In my book "NMR" is both solid-state and liquid-state. Solid-state often includes magic angle spinning. While your method is digitally-based where many practical difficulties are non-existent (as it would be for a parallel MAS setup) I would still imagine that a digital high throughput twin for par-MAS-ssNMR would be

challenging. If you agree, please limit your claim to what you have shown: few-channel par-lsNMR. Use your stress testing on number of channels if you desire to expand your claim.

We agree that this work does not specifically address practical challenges encountered in solid-state NMR. It is essential to note that we do not impose restrictions on the number of channels in the digital twin. For demonstration, we have implemented a 16-channel setup in simulations and a 2-channel experiment with two water samples as well as two samples of L-alanine and L-valine.

Increasing the number of channels leads to a larger computational mesh size, elevated degrees of freedom in electromagnetic simulations, and a larger scale Hamiltonian in spin dynamics calculations. These factors necessitate enhanced computational capabilities. The wall-clock time results indicate that a working PC can handle a 16-channel simulation, requiring approximately 3.5 hours. Notably, the optimal control operates on a single channel, rendering its computational effort independent of the degree of detector parallelism. Furthermore, the BSS-based signal decomposition now supports the identification of hundreds of sources [6]. These facts enhance the generality and utility of the digital twin for parallel NMR simulations.

In conclusion, the modified title now reads as "A digital twin for parallel liquid-state NMR spectroscopy".

References

- [1] Rasulov, U., Acharya, A., Carravetta, M., Mathies, G. & Kuprov, I. Simulation and design of shaped pulses beyond the piecewise-constant approximation. *Journal of Magnetic Resonance* **353**, 107478 (2023).
- [2] Tošner, Z. *et al.* Overcoming volume selectivity of dipolar recoupling in biological solid-state NMR spectroscopy. *Angewandte Chemie International Edition* **57**, 14514–14518 (2018).
- [3] Goodwin, D. L. & Vinding, M. S. Accelerated Newton-Raphson GRAPE methods for optimal control. *Physical Review Research* **5**, L012042 (2023).

- [4] Wang, H., Ciobanu, L., Edison, A. S. & Webb, A. G. An eight-coil high-frequency probehead design for high-throughput nuclear magnetic resonance spectroscopy. *Journal of Magnetic Resonance* **170**, 206–12 (2004).
- [5] Cheng, Y.-T., Jouda, M. & Korvink, J. Sample-centred shimming enables independent parallel NMR detection. *Scientific Reports* **12**, 14149 (2022).
- [6] Kervazo, C., Bobin, J. & Chenot, C. Blind separation of a large number of sparse sources. *Signal Processing* **150**, 157–165 (2018).

Reviewers' comments:

Reviewer #1 (Remarks to the Author):

The authors have carefully revised the manuscript based on the first round of reviews. The addition of a significant amount of detail to the "supplementary information" document is particularly appreciated.

The authors have adequately addressed my concerns. I have no further comments.

Reviewer #2 (Remarks to the Author):

I really thank the Authors for their efforts. The revised version is greatly improved and many technical, but important, steps are discussed in detail.

In the following some minor points to address:

1) The "The simulation provides an unloaded Sc matrix, ..." (Revised article Page 5) and the S2 section states that "Note that applying optimization needs the unload S matrix information". Now the load changes the Sc matrix and the optimal tuning/matching capacitors as well. I missed in which step the Authors fully consider the inclusion of the sample since, reading the manuscript/supplementary I got the impression that Sc refers to unloaded coils as well Scm, the F and G matrices.

2) Page 14, first paragraph of Discussion, "filed" should be "field"

3) In Reference 29 the authors are reported with explicit names instead of surnames

4) Supp. Page 4.

- "tunning" should be "tuning".

- Please revise the following text: "When the coupling increases and results in an unacceptable mismatch, for example, $SCM_{11} > -20$ dB, as shown in Fig. S4(a). The capacitance parameters are then optimized to minimize the diagonal elements of SCM."

- Please clarify if the optimization performed "to minimize the diagonal elements of SCM" is done using different Ct and Cm capacitors for each channel.

5) Supp. Page 7, please revise the sentence that contains eq. (S11)

6) Supp. S5, the symbol “ δ ” is used for a random number while in S15 it is declared as the compensation pulse term. This can confuse the readers.

Nice piece of work!

Reviewer #3 (Remarks to the Author):

I thank the authors for answers to my inputs. You have chosen to downsize your claims rather than pursue them. This is in order, but I had hoped you stepped up instead. I think the computation times involved for more and more channels raise a red flag. From that I can understand why you did not cite more OC papers from the MRI field which don't suffer from those computation times after all, even with 16 channels, no coupling matrices etc. It would be in order to mention that if one pursues the MRI way of exciting spins with multiple channels, there are multi-channel OC solutions for this. It was hinted but I think it could be more stressed.

From my comments and the other reviewers', which you have addressed, I think the weight of knowledge between the main text and the supplementary material is too skewed. Many of the important concerns raised are mainly addressed in the supplementary, where they are given less importance so to speak. The main text is definitely not a read-alone document. It comes with a long-read supplementary. This is both good and bad, but I think a paper is better if it can be read 90% on its own. Basically, I think you could have revised the main text more, although I recognize this is indeed a challenge. You cover a lot ground in one paper. I will let the Editor decide.

Response to Reviewers

April 9, 2024

We sincerely appreciate the reviewers for reading the revised paper and providing positive feedback. Below, we present a detailed response to the comments raised by Reviewers 2 and 3. Our responses and all modifications to our article are highlighted in blue.

Reviewer 1

The authors have carefully revised the manuscript based on the first round of reviews. The addition of a significant amount of detail to the "supplementary information" document is particularly appreciated.

The authors have adequately addressed my concerns. I have no further comments.

We're grateful for the reviewer's positive feedback and helpful input in refining the manuscript.

Reviewer 2

I really thank the Authors for their efforts. The revised version is greatly improved and many technical, but important, steps are discussed in detail.

We appreciate the reviewer's evaluation of the quality improvement. The minor comments will further polish our work.

In the following some minor points to address:

1) The "The simulation provides an unloaded Sc matrix, . . ." (Revised article Page 5) and the S2 section states that "Note that applying optimization needs the unload S matrix information". Now the load changes the Sc matrix and the optimal tuning/matching capacitors as well. I missed in which step the Authors fully consider the inclusion of the sample since, reading the manuscript/supplementary I got the impression that Sc refers to unloaded coils as well Scm, the F and G matrices.

Thank you for bringing it to our attention. It was indeed an inappropriate description. The sample was included in simulating the Sc matrix, as depicted in Fig. S6(a) in the

Supplementary, we also added this coil geometry to Fig. 2(b) for illustration. The term 'unload Sc matrix' has been corrected to 'coil array S matrix'. The subsequent RF modeling was based on this matrix.

2) Page 14, first paragraph of Discussion, "filed" should be "field"

Corrected, thank you.

3) In Reference 29 the authors are reported with explicit names instead of surnames

Corrected, thank you for reminding.

4) Supp. Page 4. - "tunning" should be "tuning".

Corrected, thank you.

- Please revise the following text: "When the coupling increases and results in an unacceptable mismatch, for example, $S_{CM,11} > -20dB$, as shown in Fig. S4(a). The capacitance parameters are then optimized to minimize the diagonal elements of SCM."

The sentence has been revised and flows better now. Thank you.

- Please clarify if the optimization performed "to minimize the diagonal elements of SCM" is done using different Ct and Cm capacitors for each channel.

Interesting question. By using different tuning and matching capacitors for each channel, we tested the 2-channel case (Fig. R1) and 16-channel case (Fig. R2). In the 2-channel case, as coupling strengthens, $\|G - F\|$ tends to increase, but remains on the order of 10^{-3} . Additional fluctuations arise due to the stochastic nature of the solution. However, the 16-channel case presents an optimization challenge. With a larger parameter space, optimization may converge to local minima. When tuning an NMR coil array with symmetric geometry, it's reasonable to select identical capacitor sets for each channel.

Figure R1 Tuning and matching for 2-channel coil array. (a) The reflection at the 2 ports. (b) The distance between normalized G and F , i.e., $\|G - F\|$.

Figure R2 Tuning and matching for 16-channel coil array, (a) The convergence curve. (b) The reflection and coupling results.

5) Supp. Page 7, please revise the sentence that contains eq. (S11)

Updated, thank you.

6) Supp. S5, the symbol “ δ ” is used for a random number while in S15 it is declared as the compensation pulse term. This can confuse the readers.

Thanks for your reminding, the δ in supplementary section S5 has been replaced with χ to avoid confusion.

Nice piece of work!

Reviewer 3

I thank the authors for answers to my inputs. You have chosen to downsize your claims rather than pursue them. This is in order, but I had hoped you stepped up instead. I think the computation times involved for more and more channels raise a red flag. From that I can understand why you did not cite more OC papers from the MRI field which don't suffer from those computation times after all, even with 16 channels, no coupling matrices etc. It would be in order to mention that if one pursues the MRI way of exciting spins with multiple channels, there are multi-channel OC solutions for this. It was hinted but I think it could be more stressed.

We agree that the proposed decoupling scheme is suitable for situations where the B_1 field is confined to the coils. However, in scenarios with strong coupling, pulse compensation based solely on the coupling matrix may not suffice. Therefore, exploring simultaneous optimization schemes, as extensively addressed in MRI, could prove beneficial. We included this consideration in the discussion section.

From my comments and the other reviewers', which you have addressed, I think the weight of knowledge between the main text and the supplementary material is too skewed. Many of the important concerns raised are mainly addressed in the supplementary, where they are given less importance so to speak. The main text is definitely not a read-alone document. It comes with a long-read supplementary. This is both good and bad, but I think a paper is better if it can be read 90% on its own. Basically, I think you could have revised the main text more, although I recognize this is indeed a challenge. You cover a lot ground in one paper. I will let the Editor decide.

Thank you for your kind consideration. We understand how vital it is to keep a paper readable and concise.

We have integrated important elements including the end-to-end test results, wall-clock-time table, and figure of merit of signal decomposition into the main text. This serves to highlight the robustness and time complexity of the digital twin, as well as explain the effectiveness of the decomposition method in dense arrays.

In the main text, We removed unnecessary references to supplementary. In the residual 3

references providing the performance data of cooperative pulse and signal decomposition, we gave concise summary sentences for easy comprehension. We are confident that the main text can now stand alone as a complete article.